# Offline Guarded Safe Reinforcement Learning for Medical Treatment Optimization Strategies

**Runze Yan**[1]* **Xun Shen**[2]* **Akifumi Wachi**[3] **Sebastien Gros**[4] **Anni Zhao**[1] **Xiao Hu**[1]

[1]Emory University, [2]Tokyo University of Agriculture and Technology,
[3]LY Corporation, [4]Norwegian University of Science and Technology
runze.yan@emory.edu, shen@go.tuat.ac.jp

## Abstract

When applying offline reinforcement learning (RL) in healthcare scenarios, the out-of-distribution (OOD) issues pose significant risks, as inappropriate generalization beyond clinical expertise can result in potentially harmful recommendations. While existing methods like conservative Q-learning (CQL) attempt to address the OOD issue, their effectiveness is limited by only constraining action selection by suppressing uncertain actions. This action-only regularization imitates clinician actions that prioritize short-term rewards, but it fails to regulate downstream state trajectories, thereby limiting the discovery of improved long-term treatment strategies. To safely improve policy beyond clinician recommendations while ensuring that state-action trajectories remain in-distribution, we propose *Offline Guarded Safe Reinforcement Learning* (OGSRL), a theoretically grounded model-based offline RL framework. OGSRL introduces a novel dual constraint mechanism for improving policy with reliability and safety. First, the OOD guardian is established to specify clinically validated regions for safe policy exploration. By constraining optimization within these regions, it enables the reliable exploration of treatment strategies that outperform clinician behavior by leveraging the full patient state history, without drifting into unsupported state-action trajectories. Second, we introduce a safety cost constraint that encodes medical knowledge about physiological safety boundaries, providing domain-specific safeguards even in areas where training data might contain potentially unsafe interventions. Notably, we provide theoretical guarantees on safety and near-optimality: policies that satisfy these constraints remain in safe and reliable regions and achieve performance close to the best possible policy supported by the data. When evaluated on the MIMIC-III sepsis treatment dataset, OGSRL demonstrated significantly better OOD handling than baselines. OGSRL achieved a 78% reduction in mortality estimates and a 51% increase in reward compared to clinician decisions.

## 1 Introduction

Deep reinforcement learning (RL) has been widely applied in many safety-critical domains, such as fine-tuning of language models [9, 35], robotics [7, 8], and autonomous driving [21]. Given its capacity to learn from large-scale real-world datasets, there is growing interest in leveraging deep RL for decision support in medical treatment. Notably, deep RL has been explored for treatment optimization in various clinical conditions, including sepsis [22, 37], cancer [47], and type 2 diabetes [56]. In medical applications, unlike conventional RL, two additional challenges must be addressed. First, medical treatment optimization is not amenable to learning via active interaction; that is, online exploration of treatment alternatives for patients is strictly prohibited. Second, medical treatments are

---

*R. Yan and X. Shen contributed equally to this work.

39th Conference on Neural Information Processing Systems (NeurIPS 2025).

multi-faceted: we need to incorporate (possibly conflicting) safety constraints and a reward function. Even if a treatment is highly effective, therapies with severe side effects are undesirable for patients.

Offline RL learns policies from pre-collected datasets without further environment interaction [28], making it ideal for medical treatment optimization, where real-time experimentation is ethically constrained. Early healthcare applications relied on value-based off-policy methods such as DQN [22, 37, 54] and its variants [14, 17, 39, 44, 53]. They face challenges in offline settings due to OOD actions [25] and Q-value overestimation for unseen actions [2], leading to unsafe or suboptimal decisions. Conservative Q-learning (CQL) [26] mitigates OOD action overestimation by penalizing value estimates for actions not present in the dataset and has been applied to clinical decision-making [11, 33]. CQL focuses solely on suppressing OOD actions, leaving OOD states unaddressed [31]. As policies evolve, even in-distribution actions can lead to state trajectories that diverge from data distribution. This is problematic in healthcare, where accurate modeling of state transitions is critical, and OOD states may correspond to unsafe or clinically invalid patient conditions. Prior methods fail to fully leverage clinician expertise embedded in dataset. CQL can only encourage policies that imitate clinician actions but cannot improve upon them because it lacks mechanisms to safely explore or optimize within the full state-action support derived from expert trajectories.

**Contributions.** We propose *Offline Guarded Safe Reinforcement Learning* (OGSRL), a theoretically grounded framework for learning safe and effective treatment policies from offline clinical data. Our key contributions are as follows. (1) We introduce an OOD guardian that jointly constrains policies to remain within the state-action support and enables optimization within this region. Unlike prior methods such as CQL that only suppress OOD actions, OGSRL explicitly restricts both states and actions, fully leveraging clinician knowledge embedded in the dataset and incorporating explicit safety cost constraints to avoid risky recommendations. (2) We provide theoretical guarantees that any policy satisfying the OOD cost constraint remains in-distribution with high probability. When combined with model-based RL, OGSRL further offers probabilistic guarantees on safety and near-optimality, and quantifies the effect of dataset size on policy reliability. (3) We demonstrate the practical effectiveness of OGSRL on real-world sepsis treatment data. OGSRL consistently outperforms strong offline RL baselines in cumulative reward, safety constraint satisfaction, and alignment with clinical behavior.

## 2 Problem Statement

**Modeling medical treatment as a CMDP.** We define the patient state as $\mathbf{s} \in \mathcal{S} \subseteq \mathbb{R}^n$ and the permissible treatment action as $\mathbf{a} \in \mathcal{A} \subseteq \mathbb{R}^m$. The patient state evolves according to a transition dynamics $\mathcal{T}(\mathbf{s}^+ \mid \mathbf{s}, \mathbf{a})$, which specifies the distribution over the next state $\mathbf{s}^+$ given the current state $\mathbf{s}$ and action $\mathbf{a}$. A reward function $r : \mathcal{S} \times \mathcal{A} \to [0, r_{\max}]$ is defined based on clinical health indicators, which reflects the treatment objective of improving patient health. In addition to reward, certain safety indicators must be considered during treatment. These are encoded by a vector-valued safety cost function $\mathbf{c} : \mathcal{S} \times \mathcal{A} \to [0, c_{1,\max}] \times \cdots \times [0, c_{\ell,\max}]$, where $\mathbf{c}_{\max} = [c_{1,\max}, \ldots, c_{\ell,\max}]^\top$ denotes the upper bounds for $\ell$ safety-related quantities. At each decision step $h$, a clinician observes the current patient state $\mathbf{s}_h$ and selects a treatment $\mathbf{a}_h$ aimed at improving the patient's condition (maximizing $r$) while avoiding unsafe outcomes (ensuring each component of $\mathbf{c}$ remains within safe limits). Thus, medical treatment can be formulated as a *constrained Markov decision process* (CMDP) by $\mathcal{M} := \langle \mathcal{S}, \mathcal{A}, \mathcal{T}, r, \mathbf{c}, \gamma, \rho_0 \rangle$, where $\gamma \in (0, 1]$ is a discount factor and $\rho_0$ is the probability density of the initial patient state $\mathbf{s}_0$, typically reflecting the variety of conditions at the time of ICU admission or treatment onset. A treatment policy is a stochastic mapping from the state to the probability density over admissible treatment actions. Let $\pi(\cdot \mid \mathbf{s})$ denote the probability density of $\mathbf{a}$ when the state is $\mathbf{s}$, and let $\Pi$ be the space of all such policies. Let $\tau := \{\mathbf{s}_0, \mathbf{a}_0, \ldots, \mathbf{s}_h, \mathbf{a}_h, \ldots\}$ represent a trajectory induced by a policy $\pi \in \Pi$. The value function associated with a bounded function $\diamond : \mathcal{S} \times \mathcal{A} \to \mathbb{R}$ (e.g., reward $r$ or a safety cost component $c_j$) under policy $\pi$ and transition dynamics $\mathcal{T}$ is defined by $V_{\diamond, \mathcal{T}}^\pi(\mathbf{s}) = \mathbb{E}_\pi[\sum_{h=0}^\infty \gamma^h \diamond (\mathbf{s}_h, \mathbf{a}_h) \mid \mathbf{s}_0 = \mathbf{s}]$. Here, $\diamond$ is assumed to be bounded by $\diamond_{\max}$. The expected value across patients is defined as $V_{\diamond, \mathcal{T}}^\pi(\rho_0) := \mathbb{E}_{\mathbf{s} \sim \rho_0}[V_{\diamond, \mathcal{T}}^\pi(\mathbf{s})]$.

**Example scenario.** Consider the treatment of sepsis. Conventional studies of using RL to optimize sepsis treatment have not considered safety constraints either for the action or for physiological states that a safe treatment action should always maintain. Early studies [22, 36, 45] of sepsis treatment used mortality as the only penalty (negative reward) to guide the learning process, but recent work [14, 15, 19, 53] started using composite scores, such as the Sequential Organ Failure Assessment (SOFA), as the negative or the reciprocal of the reward. While SOFA combines multiple organ function

---

**Algorithm 1** OGSRL: Offline Guarded Safe Reinforcement Learning for Treatment Recommendation

---

1: **Input** Initial dataset $\mathcal{D}_\mathsf{b}$ collected under standard treatment
2:   Learn classifier $\hat{g}$ of the guardian from $\mathcal{D}_\mathsf{b}$ to detect safe state-action pairs (see Sec. 3.1)
3:   Construct guarded treatment model $\widehat{\mathcal{M}}_{\hat{g}}$ using $\hat{g}$ and $\mathcal{D}_\mathsf{b}$ (see Def. 2)
4:   $\hat{\pi} \leftarrow \mathsf{ConOpt}(\widehat{\mathcal{M}}_{\hat{g}})$ to compute a safe and effective treatment policy
5: **end for**

---

indicators and hence encourages actions that move a patient towards normal physiological states, the learning algorithm cannot guarantee that every intermittent physiological state of a patient is indeed safe. In addition, there are readily available variables that are not part of SOFA but can be used to produce physiologically sound and clinically interpretable safety constraints. Hence, a novel algorithm that is capable of learning policies that explicitly obey safety constraints is needed.

**Goal.** The primary objective of this paper is to maximize the value function $V_{r,\mathcal{T}}^\pi(\rho_0)$, while ensuring that the adopted treatment policy $\pi$ should satisfy the safety cost constraints: $V_{c_j,\mathcal{T}}^\pi(\rho_0) \le \bar{c}_j,\ j \in [\ell]$, where $\bar{c}_j \in [0, c_{j,\max}]$ is the upper constraint for the $j$-th expected cumulative safety cost. The safe RL problem associated with $\rho_0$ we shall solve is written as

$$\max_{\pi \in \Pi} V_{r,\mathcal{T}}^\pi(\rho_0) \quad \text{s.t.} \quad V_{c_j,\mathcal{T}}^\pi(\rho_0) \le \bar{c}_j, \quad \forall j \in [\ell]. \tag{ESRL}$$

## 3   Method

We propose a framework called Offline Guarded Safe Reinforcement Learning (OGSRL) to learn a treatment policy under safety constraints. The workflow of OGSRL is outlined in Algorithm 1. First, the offline dataset $\mathcal{D}_\mathsf{b} := \{(\mathbf{s}, \mathbf{a}, \mathbf{s}_+, r, \mathbf{c})\}$ is used to estimate the reward function, safety cost function, and transition dynamics, which together define an *estimated constrained Markov decision process (E-CMDP)*. To address the risk of unsafe generalization, we incorporate a *guardian* into the model-based offline safe RL and construct a *guarded E-CMDP*. The guardian plays two roles: classification and rejection. A PSoS-based classifier $\hat{g}$ is trained to identify OOD state-action pairs. Using the learned classifier $\hat{g}$, we formulate an OOD cost constraint and insert it into the CMDP. The OOD cost constraint explicitly eliminates policies with a high probability of visiting state-action pairs outside the dataset support. Unlike CQL that primarily suppresses OOD actions, our constraint jointly addresses both OOD states and actions, leading to improved generalization and safety. A constrained policy optimizer, denoted as ConOpt, is then used to solve the guarded E-CMDP and compute a policy $\hat{\pi}^{(i)}$ that maximizes the expected clinical outcome while satisfying the predefined safety constraints and additional OOD constraint. While we employed constrained policy optimization (CPO, [1]) as ConOpt, other constrained RL algorithms are not prohibited from being used.

### 3.1   Constructing Guardian Classifier

The state-action space $\mathcal{U} := \mathcal{S} \times \mathcal{A}$ can be partitioned into two regions: the in-distribution (ID) set $\mathcal{U}_\mathsf{id}$ and the OOD set $\mathcal{U}_\mathsf{ood} := \mathcal{U} \setminus \mathcal{U}_\mathsf{id}$. The estimated model is only guaranteed to converge in the ID region $\mathcal{U}_\mathsf{id}$. To prevent unsafe generalization, we introduce a *guardian* that classifies whether a state-action pair lies outside the support of the data and then restricts policy learning to ID regions.

We first introduce an important notion called polynomial sublevel set, defined as follows.

**Definition 1** (Polynomial sublevel set). *Let* $\mathbf{x} = (\mathbf{s}, \mathbf{a}) \in \mathbb{R}^{n_\mathsf{p}}$ *with* $n_\mathsf{p} = n + m$. *Let* $\mathbf{e}(\mathbf{x})$ *denote the vector of all monomials of* $\mathbf{x}$ *up to degree* $d > 0$, $\mathbf{e}(\mathbf{x}) := [1, x_1, \ldots, x_{n_\mathsf{p}}, x_1^2, \ldots, x_{n_\mathsf{p}}^d]^\top$. *Given parameter vector* $\theta$, *define the polynomial function:* $q(\mathbf{x}, \theta) := \mathbf{e}^\top(\mathbf{x}) P(\theta) \mathbf{e}(\mathbf{x})$, *where* $P(\theta)$ *is a symmetric, positive semidefinite Gram matrix fully determined by* $\theta$. *The degree of* $q$ *is* $2d$, *and we require* $q(\mathbf{x}, \theta) \ge 0$ *for all* $\mathbf{x}$, *making it a polynomial sum-of-squares (SoS) function [24, 41, 42]. Then, the polynomial sublevel set is given by:* $\widehat{\mathcal{U}}_{\theta,d} := \{\mathbf{x} \in \mathcal{U} : q(\mathbf{x}, \theta) \le 1\}$.

Ideally, we desire to obtain the following classifier $g : \mathcal{S} \times \mathcal{A} \to \{0, 1\}$, defined as $g(\mathbf{s}, \mathbf{a}) = \mathbb{I}\{(\mathbf{s}, \mathbf{a}) \notin \mathcal{U}_\mathsf{id}\}$. Unfortunately, the perfect classifier $g$ is unknown in practice. We thus aim to approximate this set using a polynomial sum-of-squares (PSoS) classifier, which enables explicit theoretical analysis of the OOD guarantee due to its structured mathematical form. While PSoS

provides analytic tractability for safety proofs, we use a kernel-based approximation in practice to improve scalability and ease of implementation. As such, by learning a polynomial sublevel set $\widehat{\mathcal{U}}_{\theta,d}$ satisfying $\widehat{\mathcal{U}}_{\theta,d} \subseteq \mathcal{U}_{\mathsf{id}}$ with high probability, we obtain a conservatively approximated classifier, denoted as $\hat{g} : \mathcal{S} \times \mathcal{A} \to \{0, 1\}$: $\hat{g}(\mathbf{s}, \mathbf{a}) = \mathbb{I}\left\{(\mathbf{s}, \mathbf{a}) \notin \widehat{\mathcal{U}}_{\theta,d}\right\}$, where $\widehat{\mathcal{U}}_{\theta,d}$ is a degree-$d$ polynomial sublevel set parameterized by $\theta \in \mathbb{R}^{n_\theta}$. Learning the PSoS guardian $\hat{g}$ involves estimating the polynomial sublevel set $\widehat{\mathcal{U}}_{\theta,d}$ from the dataset $\mathcal{D}_{\mathsf{b}}$. Let $\mathcal{X}_N := \left\{\mathbf{x}^{(i)}\right\}_{i=1}^{N}$ denote the collection of $N$ state-action pairs sampled from $\mathcal{D}_{\mathsf{b}}$. Optimization problem for constructing $\widehat{\mathcal{U}}_{\theta,d}$ is given by:

$$\min_{\theta} \quad L(\theta) := \log \det P^{-1}(\theta) \quad \text{s.t.} \quad \frac{1}{N} \sum_{i=1}^{N} \mathbb{I}_1\left(q(\mathbf{x}^{(i)}, \theta)\right) \leq \alpha_{\mathsf{c}}, \quad \text{(GCL)}$$

where $\alpha_{\mathsf{c}} \in (0, 1)$ is an empirical coverage threshold and $\mathbb{I}_1(z) = 1$ if $z > 1$, and 0 otherwise. The objective minimizes the volume of the set, forming a tight envelope around the in-support data. This set is later used to detect whether a state-action pair is out-of-distribution. In practice, $\mathbb{I}_1$ is replaced with a smooth surrogate for tractability. While we adopt this PSoS-based classifier for theoretical guarantees, alternative methods such as Kernel Density Estimation (KDE) [16] or $k$-Nearest-Neighbors ($k$-NN) scoring [6] can approximate the support and are used in our experiments (Appendix G.3). Let $\hat{\theta}_{\alpha_{\mathsf{c}}}^{N}$ denote the solution to this problem, and define the learned set as $\widehat{\mathcal{U}}_{\hat{\theta}_{\alpha_{\mathsf{c}}}^{N},d}$.

**Probability bound of guardian classifier learning.** In medical applications, it is particularly important to use an algorithm with favorable theoretical properties. We now provide a probabilistic guarantee on the accuracy of the learned classifier used in the guardian.

**Theorem 1.** *For any probability level $\alpha > 0$ and any $\alpha_{\mathsf{c}} > \alpha$, there exists a polynomial degree $d$ such that the following holds:* $\mathsf{Pr}\left(\widehat{\mathcal{U}}_{\hat{\theta}_{\alpha_{\mathsf{c}}}^{N},d} \not\subset \mathcal{U}_{\mathsf{id}}\right) \leq \exp\left(-2N^2(\alpha_{\mathsf{c}} - \alpha)\right)$. *That is, with high probability, all points within $\widehat{\mathcal{U}}_{\hat{\theta}_{\alpha_{\mathsf{c}}}^{N},d}$ lie in the in-distribution region $\mathcal{U}_{\mathsf{id}}$.*

The proof of Theorem 1 is provided in Appendix C. Theorem 1 implies that the learned guardian classifier provides a high-confidence rejection region, whose conservativeness is explicitly tunable via $\alpha_{\mathsf{c}}$ and improves with more data. Although our proposed method embeds the guardian classifier into the model-based offline RL, it can also be applied to model-free offline RL.

## 3.2 Model-based Offline RL with Guardian

**Model-based offline RL.** In our model-based reinforcement learning framework, we first estimate the following from the offline dataset $\mathcal{D}_{\mathsf{b}}$: a reward model $\hat{r}$, a vector-valued safety cost model $\hat{\mathbf{c}}$, and a transition dynamics model $\widehat{\mathcal{T}}$. These models define what we refer to as an *Estimated Constrained Markov Decision Process (E-CMDP)*: $\widehat{\mathcal{M}} := \langle \mathcal{S}, \mathcal{A}, \widehat{\mathcal{T}}, \hat{r}, \hat{\mathbf{c}}, \rho_0 \rangle$. Given $\widehat{\mathcal{M}}$, the model-based safe reinforcement learning problem is formulated as:

$$\max_{\pi \in \Pi} \quad V_{\hat{r},\widehat{\mathcal{T}}}^{\pi}(\rho_0) \quad \text{s.t.} \quad V_{\hat{c}_j,\widehat{\mathcal{T}}}^{\pi}(\rho_0) \leq \bar{c}_j, \quad \forall j \in [\ell], \quad \text{(MSRL)}$$

where $V_{\hat{r},\widehat{\mathcal{T}}}^{\pi}(\rho_0)$ denotes the expected clinical outcome (e.g., improvement in SOFA score), and each constraint ensures that the expected safety-related cost (e.g., risk of hypotension, organ failure, etc.) remains below a clinically acceptable threshold $\bar{c}_j$. Problem MSRL differs from the ideal formulation using the true CMDP $\mathcal{M}$, because it relies entirely on estimated models. In practice, the reward and cost functions can be learned using Gaussian process regression (GPR) [48, 49], while the transition dynamics can be estimated using techniques such as, e.g., Gaussian process models [13], or generative models [43]. A critical challenge in medical applications is that the offline dataset often covers only a limited subset of the state-action space—i.e., treatments observed under the standard of care [52]. Consequently, the learned policies are reliable only within the distribution of data induced by the behavior policy. Naively applying constrained policy optimization to this E-CMDP can result in over-optimistic value estimates and unsafe treatment decisions, especially in regions not well-covered by the data [30, 31]. To address this, we introduce a *state-action guardian* in the next step.

**Guarded E-CMDP.** With the learned PSoS classifier $\hat{g}$, we define a *guarded E-CMDP* by embedding the OOD-aware safety mechanism directly into the model:

**Definition 2.** *A guarded E-CMDP is defined as $\widehat{\mathcal{M}}_{\hat{g}} := \langle \mathcal{S}, \mathcal{A}, \widehat{\mathcal{T}}, \hat{r}, \hat{\mathbf{c}}, \rho_0, \hat{g}, \bar{c}_{\hat{g}} \rangle$, where $\bar{c}_{\hat{g}}$ is a threshold limiting the out-of-distribution (OOD) cost. The OOD cost constraint is formulated as:*

$$V_{\hat{g},\widehat{\mathcal{T}}}^{\pi}(\rho_0) := \mathbb{E}_{\mathbf{s}\sim\rho_0}\left[ V_{\hat{g},\widehat{\mathcal{T}}}^{\pi}(\mathbf{s}) \right] \leq \bar{c}_{\hat{g}}. \tag{1}$$

Given this structure, the guarded policy optimization problem is formulated as:

$$\max_{\pi\in\Pi} \quad V_{\hat{r},\widehat{\mathcal{T}}}^{\pi}(\rho_0) \text{ s.t. } \quad V_{\hat{g},\widehat{\mathcal{T}}}^{\pi}(\rho_0) \leq \bar{c}_{\hat{g}}, \ V_{\hat{c}_j,\widehat{\mathcal{T}}}^{\pi}(\rho_0) \leq \bar{c}_j, \quad \forall j \in [\ell]. \tag{GSRL}$$

The motivation for introducing the OOD cost constraint (1) is to discourage policies that frequently visit state-action pairs outside the support of the dataset. When the support of the true transition dynamics is unbounded, it is often impractical to enforce strict avoidance of OOD state-action pairs. Instead, a more tractable goal is to ensure that the policy remains within the data support with high probability over a finite horizon, formalized as the following joint chance constraint: $\Pr\{\hat{g}(\mathbf{s}_h, \mathbf{a}_h) = 0, \ \forall h \leq H\} > 1 - \beta$. However, directly incorporating this joint chance constraint into policy optimization is intractable in most safe RL frameworks. Following the approach of Shen et al. [40], we approximate it conservatively via the OOD cost constraint (1). The key idea is that, for a given risk level $\beta$, one can select a sufficiently large discount factor $\gamma$ so that feasibility under the cost constraint implies feasibility under the joint chance constraint. A discussion of this approximation strategy and practical guidance on choosing $\gamma$ is provided in Appendix B.

With the above notations, we extend the result of Theorem 1 into a policy-level guarantee:

**Corollary 1.** *Let $\hat{\pi}_{\mathsf{f}}$ be any feasible solution to Problem* GSRL. *Then, for a desired confidence level $\delta \in (0, 1)$, if the number of samples satisfies $N > \sqrt{\frac{\log(1/\delta)}{2(\alpha_c - \alpha)}}$, the policy $\hat{\pi}_{\mathsf{f}}$ ensures that, with probability $1 - \delta - \beta$, the agent remains within $\mathcal{U}_{\mathsf{id}}$ for all steps $h \leq H$.*

The proof of Corollary 1 is summarized in Appendix D.

**Connections to shielding methods.** Shielding methods [3, 4, 23, 32] guarantee safety during online environmental interaction by intervening when unsafe actions are about to occur. Our guardian in OGSRL shares a similar goal of constraining behavior that causes OOD issues, but operates entirely offline. Instead of correcting actions during execution, the guardian restricts the feasible policy space during offline optimization, ensuring that learned policies, with high probability, keep state-action trajectories within the dataset support over a finite horizon. Thus, while shielding ensures pointwise safety during online interactions, our approach provides probabilistic safety guarantees in the offline setting, which is crucial for medical applications where real-time corrections are infeasible.

**Practical significance.** In the context of medical treatment optimization, Theorem 1 and Corollary 1 provide essential probabilistic guarantees: only policies that maintain a high probability of remaining within the dataset support over a finite horizon $H$ are considered feasible. Crucially, any policy satisfying the OOD cost constraint operates entirely within regions where the estimated dynamics, value functions, and action-value functions are reliable. This is especially important in clinical settings, where learned policies must avoid poorly supported regions; otherwise, inaccurate modeling in such areas could lead to unsafe or ineffective treatment recommendations. Moreover, the OOD cost constraint is a data-driven proxy for clinical knowledge. Because the dataset reflects real-world clinician behavior, constraining policies to remain within support implicitly aligns the learned strategies with accepted medical practices, enhancing both interpretability and trustworthiness for deployment. However, it is important to note that clinician behavior often reflects safe individual treatment decisions, rather than globally optimal long-term strategies. Human decision-making may rely on heuristics or short-term goals, with limited integration of the patients' full historical state. A capable offline RL policy with an effective OOD cost constraint can leverage the full patient state to optimize long-term outcomes, while still adhering to the safe local actions reflected in clinical data. While methods like CQL [26] effectively suppress OOD actions, they do not constrain state transitions. This can be particularly problematic in clinical settings, where clinicians make decisions based on observed patient state trajectories. CQL lacks a mechanism to encode this temporal structure, leaving it unable to control or reason about OOD states that may emerge downstream. In contrast, our OOD guardian enables safe policy learning by jointly constraining states and actions, making it better aligned with clinical reasoning and safer for real-world deployment.

### 3.3 Safety and Sub-optimality with Finite Samples

**Value function error.** We begin by analyzing the error bound of the estimated value function associated with a function $\diamond$ (e.g., reward $r$ or safety cost $c_j$). This section assumes that the transition dynamics $\widehat{\mathcal{T}}$ are estimated using kernel density estimation (KDE). At the same time, the reward and safety cost functions are known, i.e., $\hat{\diamond} = \diamond$. This assumption is reasonable in many medical treatment settings, where both reward and safety cost functions are predefined, as is the case in our application study in Section 4. For settings where the reward and safety cost functions are unknown, we provide a generalized theoretical analysis in Appendix E, where these functions are estimated using GPR. Let $h$ be the bandwidth of the KDE, and assume that the joint density of $(\mathbf{s}^+, \mathbf{s}, \mathbf{a})$ and the marginal density of $(\mathbf{s}, \mathbf{a})$ are Hölder continuous with exponent $\zeta \in (0, 1]$.

**Theorem 2.** *Let $\pi$ be any feasible solution of Problem* GSRL. *Assume the standard KDE conditions $Nh^{n+m} \to \infty$ and $h \to 0$ as $N \to \infty$. Then, with probability at least $1 - 2\beta - 4\delta$, the following holds:* $\left| V^{\pi}_{\hat{\diamond}, \widehat{\mathcal{T}}}(\rho_0) - V^{\pi}_{\diamond, \mathcal{T}}(\rho_0) \right| \leq \varepsilon_{\mathsf{k}} + \varepsilon_H$, *where:*

$$\varepsilon_H := \frac{\gamma^{H+1}(2 - \gamma)\diamond_{\mathsf{max}}}{(1 - \gamma)^2}, \ \varepsilon_{\mathsf{k}} := \frac{\diamond_{\mathsf{max}}(\gamma - \gamma^{H+2})C_{\mathsf{den}}}{(1 - \gamma)^2} \left( h^{\zeta} + \sqrt{\frac{\log(1/\delta)}{Nh^{2n+m}}} \right).$$

*Here, $C_{\mathsf{den}}$ is a positive constant depending on the smoothness of the densities, the choice of kernel, and the dimensionality $2n + m$.*

Theorem 2 can be directly obtained from Theorem 6 in Appendix E by setting $\hat{\diamond} - \diamond = 0$ for any $(\mathbf{s}, \mathbf{a})$. This bound decomposes the total value function error into two parts; (1) $\varepsilon_{\mathsf{k}}$ from approximation of $\widehat{\mathcal{T}}$, which vanishes asymptotically; (2) $\varepsilon_H$, due to state-action pairs that fall outside the support of the dataset beyond horizon $H$. By selecting a sufficiently large dataset size $N$ and a conservative OOD threshold $\bar{c}_{\hat{g}}$, we can ensure small $\beta$ in the chance constraint (4), and thus make $\varepsilon_H$ negligible. Method of choosing $\bar{c}_{\hat{g}}$ with respect to a desired $H$ follows [40, 50].

**Safety and sub-optimality.** We now define conditions under which the policy output by ConOpt is safe and near-optimal with respect to the true model. We say a policy $\pi_{\mathsf{out}}$ is $\varepsilon_{\mathsf{s}}$-safe if: $\max_j \left| \bar{c}_j - V^{\pi_{\mathsf{out}}}_{\hat{c}_j, \widehat{\mathcal{T}}}(\rho_0) \right| \geq \varepsilon_{\mathsf{s}}$. Let $\hat{\pi}^*$ be the optimal solution to Problem GSRL with safety threshold $\bar{c}_j$. If $\pi_{\mathsf{out}}$ is computed using a tightened threshold $\bar{c}_j - \bar{\varepsilon}$, and satisfies: $V^{\hat{\pi}^*}_{\hat{r}, \widehat{\mathcal{T}}}(\rho_0) - V^{\pi_{\mathsf{out}}}_{\hat{r}, \widehat{\mathcal{T}}}(\rho_0) \leq \varepsilon_{\mathsf{r}}$, we obtain the following guarantee for the true system:

**Theorem 3.** *If $\bar{\varepsilon} \geq \varepsilon_{\mathsf{s}} + \varepsilon_{\mathsf{k}} + \varepsilon_H$, and $\pi_{\mathsf{out}}$ is $\varepsilon_{\mathsf{r}}$-sub-optimal for Problem* GSRL*, then $\pi_{\mathsf{out}}$ is safe and $(\varepsilon_{\mathsf{r}} + 2\varepsilon_{\mathsf{k}} + 2\varepsilon_H)$-sub-optimal for Problem* ESRL*, with probability at least $1 - 2\beta - 4\delta$.*

**Practical significance.** Theorem 3 guarantees that the learned policy remains safe and near-optimal with high probability, even under model approximation and conservative constraints. This is essential in clinical contexts, where decisions must be not only effective but also verifiably safe. Crucially, our approach constrains learning within the support of the dataset, where expert treatment trajectories reside, thus fully leveraging clinician knowledge while avoiding unsafe extrapolation. Unlike prior methods relying on unverifiable assumptions, our result explicitly links dataset size and model error to performance bounds, making it well-suited for reliable deployment in clinical workflows. Finally, while our approach and Off-Dynamics RL both use classifiers to influence learning, the goals differ. Our guardian is designed to restrict policy optimization to the in-distribution region for safety guarantees, whereas Off-Dynamics RL uses them for reward shaping or domain adaptation [10].

## 4 Experimental Validations

We evaluate OGSRL through comprehensive experiments on real-world clinical data to validate three key aspects of our framework: (1) the effectiveness of the OOD guardian in constraining policies to in-distribution regions, (2) the ability to learn safe and effective treatment policies that improve upon clinician behavior while satisfying physiological safety constraints, and (3) the generalizability across different critical care conditions. We conduct detailed evaluation on sepsis treatment using the MIMIC-III dataset (Sections 4.1–4.2). To demonstrate broader applicability, we validate generalizability on the Synthetic Acute Hypotension Dataset (Section 4.3), which represents a different critical care condition with distinct physiological dynamics, temporal resolution, and clinical objectives.

Across both validation studies, we instantiate OGSRL using GMB-CPO, a model-based variant of Constrained Policy Optimization equipped with our OOD guardian mechanism. [2]

## 4.1  Sepsis Treatment: Formulation and Experimental Setup

We evaluated OGSRL using $18,923$ ICU stays with sepsis diagnosis from the MIMIC-III dataset [3] [18] and established protocols in Komorowski et al. [22]. Patient data were encoded as multidimensional time series with 4-hour intervals, capturing up to 72 hours around the estimated onset of sepsis. Our implementation addresses key limitations in previous approaches to sepsis treatment optimization. Rather than discretizing interventions or combining multiple treatments into a single dimension, we developed a continuous two-dimensional action space that separately models intravenous fluid administration (IFA) and maximum vasopressor dosage (MVD), namely $\mathbf{a} = [\text{IFA}, \text{MVD}]^\top \in \mathbb{R}^2$. This representation enables more nuanced treatment recommendations, reflecting the clinical reality where physicians simultaneously titrate multiple interventions based on patient response. The state representation emerged from a clinically informed feature selection process, incorporating variables significantly correlated with organ dysfunction. This balanced representation captures essential physiological dynamics while enabling personalized treatment strategies. Totally $13$ features are selected as the dynamic state, namely, $\mathbf{s} \in \mathbb{R}^{13}$. Departing from previous work that employed mortality as a terminal reward [22], we adapted the Sequential Organ Failure Assessment (SOFA) score into an instantaneous reward signal by setting $r : \mathcal{S} \times \mathcal{A} \to \frac{1}{\text{SOFA}}$. More details about the definitions for selected dynamic and static features, actions and reward can be found in Appendix G.2. This approach provides more frequent feedback on treatment efficacy and better aligns with contemporary clinical practice. Our approach implements two distinct but complementary safety mechanisms. First, explicit safety constraints are appllied to physiological states by enforcing minimum physiological thresholds for oxygen saturation ($\text{SpO}_2$) ($\geq 92\%$) [38] and urine output ($\geq 0.5$ mL/kg/hour) [20]. These constraints directly encode clinical knowledge about vital parameter ranges necessary for patient safety. Second, our OOD guardian mechanism addresses a fundamentally different safety concern—the reliability of model predictions when encountering state-action pairs insufficiently represented in training data. While clinical constraints ensure physiological safety within the model's assumptions, the OOD guardian prevents the policy from recommending treatments in regions where the model itself may be unreliable, regardless of the predicted clinical outcomes. We implement OGSRL as described in Algorithm 1, approximating the PSoS guardian classifier $\hat{g}$ using a kernel-based method (see Appendix G.3) to identify OOD state-action pairs efficiently. For transition dynamics $\widehat{\mathcal{T}}$, we employed a $k$-nearest neighbor ($k$-NN) approach as an approximation of KDE, which maintains theoretical consistency while offering practical advantages for clinical time-series data, particularly its robustness to sparse regions in the state space [34, 51]. We use CPO [1] as the constrained policy optimizer ConOpt, resulting in our full implementation referred to as GMB-CPO, which is a model-based (MB) variant of CPO equipped with the OOD Guardian (G). Additional details are provided in Appendix G.4. Note that GMB-CPO is a specific instantiation of the proposed OGSRL framework. As discussed at the beginning of Section 3, other constrained reinforcement learning algorithms can also be employed as ConOpt within our framework.

**Baseline Algorithms and Evaluation Metrics.** We evaluated OGSRL against seven baseline algorithms spanning model-free and model-based offline RL approaches, and their guardian-enhanced variants prefixed with G: (1) CQL; (2) CQL with Guardian (GCQL); (3) CQL variant (CCQL) presented in [33]; (4) CCQL with constraint satisfaction (GCCQL); (5) MB-TRPO [29]; (6) MB-TRPO with Guardian (GMB-TRPO); (7) MB-CPO. The implementation details for guardian integration with each algorithm are summarized in Appendix G.4. We assess OGSRL against baseline methods across four critical dimensions that follow a logical progression essential for clinical deployment: (1) OOD state avoidance: establishing whether guardian mechanism effectively constrains policies to remain within the clinical data support; (2) clinical alignment: measuring how closely learned policies match clinician decision-making patterns, a prerequisite for interpretability and trust; (3) treatment effectiveness: quantifying improvements in patient outcomes compared to standard care; (4) physiological safety: verifying that policies maintain vital parameters within safe ranges throughout treatment trajectories. For quantitative evaluation, we employed four clinically relevant metrics: Model Concordance Rate (MCR) measuring alignment with clinician decisions, Appropriate Intensification Rate (AIR) assessing treatment escalation in response to physiological deterioration, Mortality

---

[2]Our source code is available at `https://github.com/Runz96/SafeRL-OGSRL`.

[3]MIMIC-III dataset: `https://physionet.org/content/mimiciii/1.4/`.

Table 1: Performance comparison across methods showing Model Concordance Rate (MCR), Appropriate Intensification Rate (AIR), Mortality Estimate (ME), and Action Change Penalties (ACP) for vasopressor dosage (MVD) and fluid administration (IFA) (mean $\pm$ Standard Deviation (SD)). Mean and SD were computed from the results of five different seeds. The symbol $\uparrow$ indicates that higher values are better, $\downarrow$ indicates that lower values are better, and $\leftrightarrow$ denotes that closer alignment with the standard of care (SOC) is preferred. MCR should align with SOC within a reasonable range.

| Method | MCR($\uparrow$, $10^{-3}$) | AIR($\uparrow$, $10^{-2}$) | ME($\downarrow$, $10^{-2}$) | ACP: MVD($\leftrightarrow$) | ACP: IFA($\leftrightarrow$, $10^2$) |
|---|---|---|---|---|---|
| CQL | $789 \pm 5.64$ | $13 \pm 0.540$ | $4.86 \pm 0.540$ | $4.18 \pm 0.129$ | $5.43 \pm 0.083$ |
| GCQL | $\mathbf{909 \pm 2.52}$ | $30.5 \pm 1.17$ | $5.53 \pm 0.214$ | $3.13 \pm 0.033$ | $1.51 \pm 0.034$ |
| CCQL | $827 \pm 3.12$ | $3.93 \pm 0.248$ | $4.81 \pm 0.339$ | $3.74 \pm 0.066$ | $4.60 \pm 0.027$ |
| GCCQL | $827 \pm 3.50$ | $30.2 \pm 0.930$ | $5.17 \pm 0.142$ | $3.23 \pm 0.110$ | $2.73 \pm 0.011$ |
| MB-TRPO | $0.04 \pm 0.055$ | $2.45 \pm 0.280$ | $-$ | $48.1 \pm 0.121$ | $1670 \pm 1.78$ |
| GMB-TRPO | $571 \pm 3.37$ | $36.9 \pm 1.19$ | $2.32 \pm 0.491$ | $1.24 \pm 0.026$ | $9.85 \pm 0.063$ |
| MB-CPO | $0.04 \pm 0.055$ | $\mathbf{49.6 \pm 0.731}$ | $-$ | $50.5 \pm 0.121$ | $492 \pm 1.12$ |
| GMB-CPO | $549 \pm 2.56$ | $\mathbf{44.8 \pm 0.241}$ | $\mathbf{1.38 \pm 0.482}$ | $\mathbf{4.34 \pm 0.052}$ | $\mathbf{6.47 \pm 0.018}$ |
| SOC | $-$ | $-$ | $\mathbf{6.32}$ | $\mathbf{4.34}$ | $\mathbf{6.48}$ |

Estimate (ME) projecting survival outcomes, and Action Change Penalty (ACP) quantifying treatment smoothness over time. Detailed definitions and computational methodology for these metrics are provided in Appendix G.6.

## 4.2 Sepsis Treatment: Results and Discussions

We present results on OOD state avoidance and summarize key insights regarding clinical efficacy, defined here as the ability of learned policies to simultaneously achieve clinician alignment, treatment effectiveness, and physiological safety. This organization allows us to focus on the core technical contribution while providing essential context on its downstream clinical implications, with comprehensive quantitative analyses available in Appendix G.7.

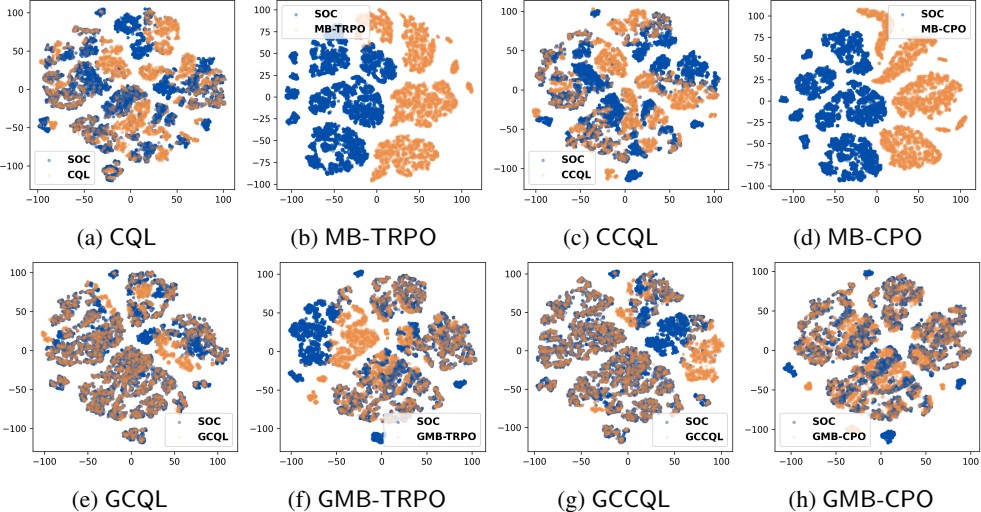

Figure 1: Results on state distributions by learned policies via different algorithms. Blue points represent the original offline dataset; orange points represent the states visited by the learned policies.

**OOD State Avoidance.** To visualize the high-dimensional state distributions, we apply t-SNE dimensionality reduction to project the policy-generated states and the original clinical dataset onto a 2D manifold. Figure 1 compares the distributions across all evaluated algorithms. Policies learned without the guardian (CQL, MB-TRPO, CCQL, MB-CPO) exhibit significant divergence from the support of the offline dataset, with many states falling outside the distribution of the training data. In contrast, guardian-augmented policies (GCQL, GMB-TRPO, GCCQL, GMB-CPO) maintain state distributions tightly concentrated around the dataset support, visually validating our theoretical guarantees on OOD state avoidance (Theorem 1 and Corollary 1). This visualization confirms the

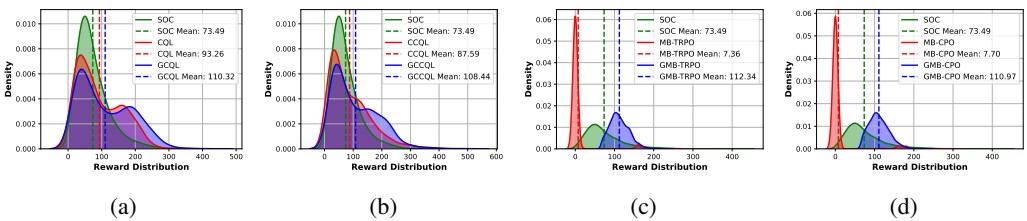

|   |   |   |   |
|---|---|---|---|
| (a) | (b) | (c) | (d) |

Figure 2: Comparison of cumulative reward distributions between the SOC (green) and policies by different algorithms with guard mechanisms (blue). Each subplot shows the estimated reward density for trajectories in the test set. Dashed vertical lines indicate the mean rewards. (a) CQL vs. GCQL;(b) CCQL vs. GCCQL; (c) MB-TRPO vs. GMB-TRPO; (d) MB-CPO vs. GMB-CPO.

core premise of our approach. Despite effective mitigation of OOD actions by CQL and CCQL, without explicit guardian mechanisms, their learned policies still induce OOD states during trajectory rollouts. Integrating the proposed guardian restricts policies to operate within regions where model predictions remain reliable, enhancing the generalization capability of existing approaches.

**Clinical Efficacy.** Table 1 presents quantitative performance metrics across all RL algorithms evaluated in our study. We observed that the incorporation of guardian mechanism led to significant performance improvements, regardless of which underlying RL algorithm was implemented. Specifically, we observed marked improvements in clinician decision alignment (MCR increased from $0.789$ to $0.909$ in GCQL and from approximately zero to $0.549$ in GMB-CPO), as did appropriate intervention timing (AIR increased from $0.130$ to $0.305$ in GCQL). These improvements directly reflect the guardian's ability to constrain policies within clinically relevant state-action regions (Corollary 1). When comparing model-free versus model-based approaches, we observed complementary strengths. Model-free methods with guardians (GCQL, GCCQL) achieved superior clinician concordance, while model-based guardian approaches (GMB-TRPO, GMB-CPO) demonstrated enhanced physiological responsiveness and more concentrated reward distributions (Figure 2), indicating greater robustness to patient variability. Notably, GMB-CPO achieved the lowest mortality estimate ($0.0138$), representing a $78.2\%$ reduction compared to the standard of care ($0.0632$), while simultaneously improving cumulative rewards by $51\%$ compared to SOC. The explicit incorporation of safety constraints on physiological states demonstrated effectiveness even without guardian integration. As evidenced by Figure 3a, MB-CPO) reduced the number of unsafe states through explicit constraints on $SpO_2$ ($\geq 92\%$) and urine output ($\geq 0.5$ mL/kg/hour), whereas MB-TRPO exhibits concerning deterioration in both physiological states. These results illustrate how explicitly encoded clinical constraints preserve physiological stability throughout treatment. Interestingly, GMB-TRPO, despite lacking explicit clinical constraints, also decreased the number of unsafe states by using only the guardian mechanism. The dual-safety mechanism in GMB-CPO, combining explicit physiological with distribution-aware guardian safety constraints, achieved the greatest decrease in unsafe states for urine output and the second-best decrease for $SpO_2$. This dual-safety mechanism further enabled GMB-CPO to maintain near-identical action smoothness to clinical practice (ACP of $4.34$ versus $4.34$ for standard care), as shown in Table 1. These improved outcomes demonstrate that GMB-CPO improves policy performance through the OOD cost constraint, consistent with Theorem 3.

### 4.3 Cross-Disease Validation: Acute Hypotension

To evaluate whether OGSRL generalizes beyond sepsis management, we validate our framework on the Synthetic Acute Hypotension Dataset [27]. This dataset represents a different critical care condition with distinct physiological dynamics and clinical objectives, providing a meaningful test of cross-disease applicability. The hypotension cohort contains 3,910 ICU stays with 187,680 hourly state-action pairs over 48 hours. Unlike sepsis experiments using 4-hour intervals with 13-dimensional states and SOFA-based rewards, hypotension operates on hourly intervals with 18-dimensional states and piecewise linear MAP-based rewards. Safety constraints also differ: urine output and lactate levels. We apply the same guardian mechanism and baseline algorithms with appropriately adjusted hyperparameters. Complete experimental setup details are provided in Appendix H.

**Results and Analysis** Table 2 summarizes performance across key metrics. The guardian mechanism demonstrates consistent benefits across both datasets. For model-free approaches, GCQL achieved

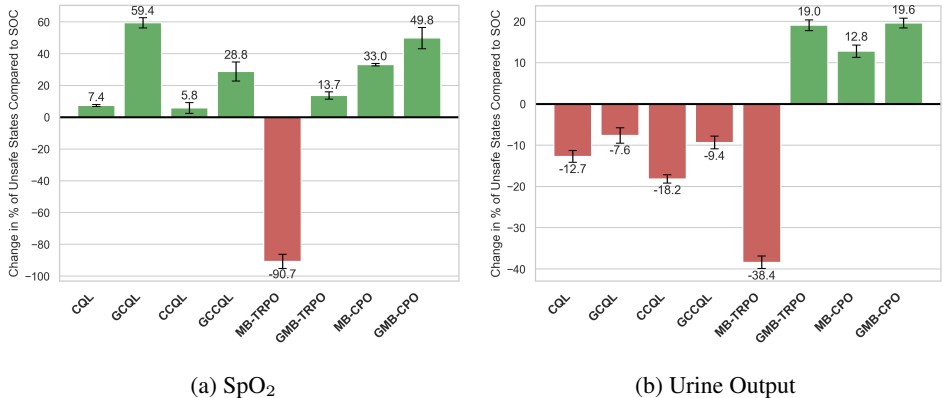

(a) SpO$_2$                    (b) Urine Output

Figure 3: Physiological safety assessment of learned policies. We evaluate the safety of learned policies by analyzing two critical physiological states: SpO$_2$ and urine output. Our assessment compares the percentage of states below defined safety thresholds against SOC. Positive values represent a reduction in unsafe states compared to SOC, while negative values an increase.

near-perfect clinician alignment (MCR: $0.973 \pm 0.002$), representing an 18% improvement over CQL ($0.824 \pm 0.004$). The impact was even more pronounced for GMB-CPO increased concordance from near-zero ($0.060 \pm 0.008$) to clinically meaningful alignment ($0.700 \pm 0.063$). In terms of clinical safety, GMB-CPO achieved the highest AIR at $0.482 \pm 0.071$—a 17% improvement over MB-CPO ($0.411 \pm 0.036$) and substantially higher than both CQL ($0.281 \pm 0.031$) and GCQL ($0.301 \pm 0.046$). Regarding cumulative rewards, GMB-CPO achieved the best performance (mean: 14.88, median: 16.14), outperforming MB-CPO (mean: 3.9, median: 5.38), GCQL (mean: 12.42, median: 12.69), and standard of care (mean: 10.37, median: 11.21).

**Cross-Disease Consistency** These results demonstrate three key aspects of cross-disease generalizability: (1) *Consistent guardian benefits*—Guardian augmentation consistently improves all methods

Table 2: Comparison on Acute Hypotension Dataset (mean $\pm$ SD).

| Method | MCR ($\uparrow$) | AIR ($\uparrow$) | Reward Mean ($\uparrow$) |
|---|---|---|---|
| CQL | $0.824 \pm 0.004$ | $0.281 \pm 0.031$ | $10.15 \pm 1.23$ |
| GCQL | $\mathbf{0.973 \pm 0.002}$ | $0.301 \pm 0.046$ | $12.42 \pm 0.98$ |
| MB-CPO | $0.060 \pm 0.008$ | $0.411 \pm 0.036$ | $3.90 \pm 2.14$ |
| GMB-CPO | $0.700 \pm 0.063$ | $\mathbf{0.482 \pm 0.071}$ | $\mathbf{14.88 \pm 1.45}$ |
| SOC | – | – | $10.37 \pm 1.87$ |

across both diseases, with model-based approaches benefiting most dramatically (concordance improved from near-zero to 0.700 for GMB-CPO in both datasets). (2) *Robust best performer*—GMB-CPO achieves the best balance of clinician alignment, clinical safety, and treatment effectiveness in both sepsis and hypotension. (3) *Mechanism transferability*—The dual-safety mechanism (explicit physiological constraints + OOD guardian) proves robust to differences in disease pathophysiology, temporal resolution (hourly vs. 4-hour intervals), state dimensionality (18 vs. 13 features), reward structure (continuous MAP-based vs. discrete SOFA-based), and safety constraints (urine + lactate vs. SpO$_2$ + urine). These cross-disease results establish OGSRL as a generalizable framework for safe offline RL in critical care settings.

## 5 Conclusion

We introduced OGSRL, a model-based offline reinforcement learning framework designed for safe and effective medical treatment optimization. By jointly enforcing OOD and safety cost constraints, OGSRL ensures policy learning remains within clinically supported regions while allowing safe performance improvement over observed clinician behavior. We established theoretical guarantees on safety, near-optimality, and in-distribution containment. We validated OGSRL by evaluating one of its instantiations, GMB-CPO, on real-world sepsis treatment data, showing substantial gains in reward, safety, and clinical consistency, demonstrating the promise of OGSRL for reliable deployment in safety-critical healthcare domains.

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

# A    Assumption on the Probability Density

Note that the data points $\{(\mathbf{s}^+, \underbrace{\mathbf{s}, \mathbf{a}}_{\mathbf{x}})\}$ in $\mathcal{D}_{\mathsf{b}}$ can be seen as samples extracted from $\mathcal{S} \times \mathcal{U}_{\mathsf{id}}$ according

to a joint probability density $f(\mathbf{s}, \mathbf{x})$ associated with the behavior policy, initial state distribution, and the transition dynamics. Here, with an abusement of notation, we replace $\mathbf{s}^+$ by $\mathbf{s}$ for simplicity. Let $p(\mathbf{s}|\mathbf{x})$ be the condition density from $f(\mathbf{s}, \mathbf{x})$. Let $f_{\mathbf{X}}(\mathbf{x})$ be the marginal density. We have the following assumption regarding the underlying real density.

**Assumption 1.** *Suppose that both joint density $f(\mathbf{s}, \mathbf{x})$ and the marginal density $f_{\mathbf{X}}(\mathbf{x})$ are Holder continuous with the parameter $\zeta \in (0, 1]$. Namely, there exists $C_\zeta$ such that $|f_{\mathbf{X}}(\mathbf{x}) - f_{\mathbf{X}}(\mathbf{x}')| \leq C_\zeta \|\mathbf{x} - \mathbf{x}'\|^\zeta$ and $|f(\mathbf{s}, \mathbf{x}) - f(\mathbf{s}', \mathbf{x}')| \leq C_\zeta \|(\mathbf{s}, \mathbf{x}) - (\mathbf{s}', \mathbf{x}')\|^\zeta$ hold. Besides, the marginal density $f_{\mathbf{X}}(\mathbf{x})$ satisfies*

$$f_{\mathbf{X}}(\mathbf{x}) \geq f_{\mathsf{min}}, \forall \mathbf{x} \in \mathcal{U}_{\mathsf{id}}. \tag{2}$$

*Both joint density and the marginal density satisfy exponential decays.*

The lower bound of the marginal density given by (2) is a strong assumption in a general sense. However, it is reasonable and practical in the problem setting of offline reinforcement learning with partial coverage. In this setting, we should not consider the area with an extremely low probability of having state-action pairs.

# B    About the Choice of $\gamma$

**Lemma 1.** *For any risk level $\beta > 0$, there exist constants $\bar{\gamma}(\beta) \in (0, 1)$ and $\bar{H}(\beta) \in \mathbb{N}$ such that, for all $\gamma \geq \bar{\gamma}(\beta)$ and $H \leq \bar{H}(\beta)$, the OOD cost constraint*

$$V_{\hat{g}, \widehat{\mathcal{T}}}^\pi(\rho_0) := \mathbb{E}_{\mathbf{s}_0 \sim \rho_0} \left[ \sum_{h=0}^\infty \gamma^h \hat{g}(\mathbf{s}_h, \mathbf{a}_h) \right] \leq \beta \tag{3}$$

*serves as a conservative approximation of the joint chance constraint:*

$$\mathsf{Pr}\{\hat{g}(\mathbf{s}_h, \mathbf{a}_h) = 0, \ \forall h \leq H\} > 1 - \beta. \tag{4}$$

*Proof.* We begin by considering the joint chance constraint under the learned dynamics model $\widehat{\mathcal{T}}$:

$$\mathsf{Pr}\left\{\hat{g}(\mathbf{s}_h, \mathbf{a}_h) = 0, \ \forall h \leq H \mid \widehat{\mathcal{T}}\right\} > 1 - \beta. \tag{5}$$

Using the definition of $\hat{g}$, this is equivalent to:

$$\mathsf{Pr}\left\{q(\mathbf{x}_h, \theta) \leq 0, \ \forall h \leq H \mid \widehat{\mathcal{T}}\right\} \geq 1 - \beta. \tag{6}$$

Define the violation probability under $\pi$ as:

$$\mathbb{V}_{\hat{g}, \widehat{\mathcal{T}}}^{H, \pi}(\rho_0) := \mathsf{Pr}\left\{\exists h \leq H : q(\mathbf{x}_h, \theta) > 0 \mid \widehat{\mathcal{T}}\right\}.$$

Then (6) is equivalent to $\mathbb{V}_{\hat{g}, \widehat{\mathcal{T}}}^{H, \pi}(\rho_0) \leq \beta$. Applying Boole's inequality gives:

$$\mathbb{V}_{\hat{g}, \widehat{\mathcal{T}}}^{H, \pi}(\rho_0) \leq \sum_{h=1}^H \mathbb{E}\left[\hat{g}(\mathbf{s}_h, \mathbf{a}_h) \mid \widehat{\mathcal{T}}\right] = \mathbb{E}\left[\sum_{h=1}^H \hat{g}(\mathbf{s}_h, \mathbf{a}_h) \mid \widehat{\mathcal{T}}\right] =: \widetilde{\mathbb{V}}_{\hat{g}, \widehat{\mathcal{T}}}^{H, \pi}(\rho_0). \tag{7}$$

Thus, $\widetilde{\mathbb{V}}_{\hat{g}, \widehat{\mathcal{T}}}^{H, \pi}(\rho_0) \leq \beta$ serves as a conservative approximation for the original joint chance constraint.

Next, consider the infinite-horizon discounted surrogate: $V_{\hat{g}, \widehat{\mathcal{T}}}^\pi(\rho_0) :=$ $\mathbb{E}_{\mathbf{s}_0 \sim \rho_0}\left[\sum_{h=0}^\infty \gamma^h \hat{g}(\mathbf{s}_h, \mathbf{a}_h) \mid \widehat{\mathcal{T}}\right]$. Let $\overline{\mathbb{V}}_{\hat{g}, \widehat{\mathcal{T}}}^{H, \pi}(\rho_0) := \mathbb{E}\left[\sum_{h=0}^H \gamma^h \hat{g}(\mathbf{s}_h, \mathbf{a}_h) \mid \widehat{\mathcal{T}}\right]$. The error between the infinite-horizon discounted cost and the cumulative cost can be expressed as:

$$\tilde{\epsilon}^\pi(\gamma, H) := V_{\hat{g}, \widehat{\mathcal{T}}}^\pi(\rho_0) - \widetilde{\mathbb{V}}_{\hat{g}, \widehat{\mathcal{T}}}^{H, \pi}(\rho_0) = V_{\hat{g}, \widehat{\mathcal{T}}}^\pi(\rho_0) - \overline{\mathbb{V}}_{\hat{g}, \widehat{\mathcal{T}}}^{H, \pi}(\rho_0) + \overline{\mathbb{V}}_{\hat{g}, \widehat{\mathcal{T}}}^{H, \pi}(\rho_0) - \widetilde{\mathbb{V}}_{\hat{g}, \widehat{\mathcal{T}}}^{H, \pi}(\rho_0)$$

$$= \underbrace{\mathbb{E}\left[\sum_{h=H+1}^\infty \gamma^h \hat{g}(\mathbf{s}_h, \mathbf{a}_h)\right]}_{\text{tail error}} + \underbrace{\mathbb{E}\left[\sum_{h=0}^H (\gamma^h - 1)\hat{g}(\mathbf{s}_h, \mathbf{a}_h)\right]}_{\text{discount bias}}.$$

This error term $\tilde{\epsilon}^\pi(\gamma, H)$ is strictly increasing in $\gamma$, with:

$$\tilde{\epsilon}^\pi(0, H) < 0, \quad \tilde{\epsilon}^\pi(1, H) > 0.$$

Therefore, for any fixed $H$ and policy $\pi$, there exists $\gamma_{\mathsf{lim}}(\pi, H)$ such that $\gamma > \gamma_{\mathsf{lim}}(\pi, H)$ implies:

$$\widetilde{\mathbb{V}}^{H,\pi}_{\hat{g},\widehat{\mathcal{T}}}(\rho_0) \leq V^\pi_{\hat{g},\widehat{\mathcal{T}}}(\rho_0).$$

If $\pi$ is parameterized over a compact set, we can define $\bar{\gamma}(H, \beta) := \sup_\pi \gamma_{\mathsf{lim}}(\pi, H)$ such that this inequality holds for all feasible policies.

We now discuss approximation under the true model $\mathcal{T}$. Let $\widetilde{\mathbb{V}}^{H,\pi}_{\hat{g},\mathcal{T}}(\rho_0)$ denote the cumulative violation cost under true dynamics. The gap between the learned-model discounted cost and the true cumulative cost is:

$$\tilde{\epsilon}^\pi_{\mathsf{s}}(\gamma, H) := V^\pi_{\hat{g},\widehat{\mathcal{T}}}(\rho_0) - \widetilde{\mathbb{V}}^{H,\pi}_{\hat{g},\mathcal{T}}(\rho_0)$$

$$\leq \underbrace{\widetilde{\mathbb{V}}^{H,\pi}_{\hat{g},\widehat{\mathcal{T}}}(\rho_0) - \widetilde{\mathbb{V}}^{H,\pi}_{\hat{g},\mathcal{T}}(\rho_0)}_{\tilde{\epsilon}^\pi_{\mathsf{s},1}} + \underbrace{V^\pi_{\hat{g},\widehat{\mathcal{T}}}(\rho_0) - \widetilde{\mathbb{V}}^{H,\pi}_{\hat{g},\widehat{\mathcal{T}}}(\rho_0)}_{\tilde{\epsilon}^\pi_{\mathsf{s},2}}.$$

The second term $\tilde{\epsilon}^\pi_{\mathsf{s},2}$ is already controlled as before. For the first term $\tilde{\epsilon}^\pi_{\mathsf{s},1}$, which reflects model mismatch, we note:

In practice, the approximation error $\tilde{\epsilon}^\pi_{\mathsf{s}}(\gamma, H)$ may be controlled by selecting a modest value of $H$ (to limit propagation of model error) and choosing $\gamma$ sufficiently close to 1 (to amplify the tail contribution in $\tilde{\epsilon}^\pi_{\mathsf{s},2}$). While this argument is heuristic, it aligns with common assumptions in model-based reinforcement learning, where shorter planning horizons and conservative discounting reduce the impact of model misspecification. Meanwhile, since the state-action pair is within the dataset support, the model misspecification can be small. Under these conditions, it is reasonable to expect $\tilde{\epsilon}^\pi_{\mathsf{s}}(\gamma, H) \geq 0$, ensuring:

$$\widetilde{\mathbb{V}}^{H,\pi}_{\hat{g},\mathcal{T}}(\rho_0) \leq V^\pi_{\hat{g},\widehat{\mathcal{T}}}(\rho_0) \leq \beta.$$

Thus, the discounted OOD cost constraint conservatively approximates the joint chance constraint under the true model. If the policy class is compact, we can define constants $\bar{\gamma}(\beta)$ and $\bar{H}(\beta)$ such that the approximation holds uniformly for all feasible policies. $\qquad\square$

## C  Proof of Theorem 1

A chance-constrained optimization problem can be formulated as

$$\min_\theta \ L(\theta) := \log \det P^{-1}(\theta) \tag{$\mathsf{P}_{\alpha,d}$}$$

$$\text{s.t.} \quad \Pr\{q(\mathbf{x}, \theta) \leq 1\} \geq 1 - \alpha. \tag{8}$$

The solution of Problem $\mathsf{P}_{\alpha,d}$ is defined by $\theta^\star_\alpha$. The corresponding polynomial sublevel set is $\widehat{\mathcal{U}}_{\theta^\star_\alpha,d}$. We have the following lemma regarding Problem $\mathsf{P}_{\alpha,d}$.

**Lemma 2.** *Assume that $\mathcal{U}_{\mathsf{id}}$ is a compact set. If $\alpha = 0$ and the degree $d \to \infty$, we have*

$$\widehat{\mathcal{U}}_{\theta^\star_\alpha,d} \to \mathcal{U}_{\mathsf{id}}. \tag{9}$$

*Proof.* Define a distance function $g_{\mathsf{s}}(\mathbf{x})$ associated with $\mathcal{U}_{\mathsf{id}}$ in the following way:

$$g_{\mathsf{s}}(\mathbf{x}) = \begin{cases} -\mathsf{dist}(\mathbf{x}, \partial\mathcal{U}_{\mathsf{id}}) + 1, & \text{if } \mathbf{x} \in \mathcal{U}_{\mathsf{id}}, \\ \mathsf{dist}(\mathbf{x}, \partial\mathcal{U}_{\mathsf{id}}) + 1, & \text{otherwise.} \end{cases} \tag{10}$$

Here, $\mathsf{dist}(\mathbf{x}, \partial\mathcal{U}_{\mathsf{id}})$ is defined by

$$\mathsf{dist}(\mathbf{x}, \partial\mathcal{U}_{\mathsf{id}}) := \inf_{\mathbf{y} \in \partial\mathcal{U}_{\mathsf{id}}} \|\mathbf{x} - \mathbf{y}\|_2. \tag{11}$$

Note that $\mathcal{U}_{\mathsf{id}}$ can be specified by $g_{\mathsf{s}}(\mathbf{x}) \leq 1$ and $g_{\mathsf{s}}(\mathbf{x})$ is continuous. By the Stone-Weierstrass theorem, for any $\varepsilon > 0$, we can find a $d$ such that the following holds:

$$\sup_{\mathbf{x} \in \mathcal{U}_{\mathsf{c}}} |g_{\mathsf{s}}(\mathbf{x}) - q^d_{\mathsf{s}}(\mathbf{x})| < \varepsilon. \tag{12}$$

Here, $\mathcal{U}_{\mathsf{c}}$ is a compact set satisfying that $\mathcal{U}_{\mathsf{id}} \subset \mathcal{U}_{\mathsf{c}}$ and $q_{\mathsf{s}}^d(\mathbf{x})$ is a $d-$degree polynomial function. Define three sets by

$$\widetilde{\mathcal{U}}_d^{\varepsilon^-} := \left\{\mathbf{x} : q_{\mathsf{s}}^d(\mathbf{x}) \leq 1 - \varepsilon\right\}, \ \widetilde{\mathcal{U}}_d := \left\{\mathbf{x} : q_{\mathsf{s}}^d(\mathbf{x}) \leq 1\right\}, \ \widetilde{\mathcal{U}}_d^{\varepsilon^+} := \left\{\mathbf{x} : q_{\mathsf{s}}^d(\mathbf{x}) \leq 1 + \varepsilon\right\}.$$

For any $\varepsilon$, let $d$ be chosen as the value that makes (12) holds. Then, we have

$$\widetilde{\mathcal{U}}_d^{\varepsilon^-} \subset \mathcal{U}_{\mathsf{id}} \subset \widetilde{\mathcal{U}}_d^{\varepsilon^+}. \tag{13}$$

Note that, as $\varepsilon \to 0$ and $d$ is corresponding chosen to satisfy (12), we have

$$\lim_{\varepsilon \to 0} \widetilde{\mathcal{U}}_d^{\varepsilon^-} = \widetilde{\mathcal{U}}_d^{\mathsf{l}}, \ \forall \varepsilon > 0, \widetilde{\mathcal{U}}_d^{\varepsilon^-} \subset \mathcal{U}_{\mathsf{id}} \Rightarrow \widetilde{\mathcal{U}}_d \subseteq \mathcal{U}_{\mathsf{id}}$$

$$\lim_{\varepsilon \to 0} \widetilde{\mathcal{U}}_d^{\varepsilon^+} = \widetilde{\mathcal{U}}_d^{\mathsf{l}}, \ \forall \varepsilon > 0, \widetilde{\mathcal{U}}_d^{\varepsilon^+} \supset \mathcal{U}_{\mathsf{id}} \Rightarrow \widetilde{\mathcal{U}}_d^{\mathsf{l}} \supseteq \mathcal{U}_{\mathsf{id}}.$$

Thus, $\widetilde{\mathcal{U}}_d^{\mathsf{l}} = \mathcal{U}_{\mathsf{id}}$.

Let $\{\varepsilon_k\}_{k=1}^{\infty}$ be a sequence converging to zero and $\{d_k\}_{k=1}^{\infty}$ be a sequence chosen to satisfy

$$\sup_{\mathbf{x} \in \mathcal{U}_{\mathsf{c}}} |g_{\mathsf{s}}(\mathbf{x}) - q_{\mathsf{s}}^{d_k}(\mathbf{x})| < \varepsilon_k, \ \forall k \in \mathbb{N}_+.$$

Define a problem for any given $k$ by

$$\min_{\theta} \ L(\theta) := \log \det P^{-1}(\theta) \tag{$\mathsf{P}_{\mathsf{r},d_k}$}$$

$$\text{s.t.} \quad q(\mathbf{x}, \theta) \leq 1, \ \forall \mathbf{x} \in \mathcal{U}_{\mathsf{id}}. \tag{14}$$

Note that Problem $\mathsf{P}_{\mathsf{r},d_k}$ is equivalent to Problem $\mathsf{P}_{\alpha,d}$ with $\alpha = 0$ and $d = d_k$. For all $k$, let $\Theta_k^{\mathsf{f}}$ be the feasible set of Problem $\mathsf{P}_{\mathsf{r},d_k}$. Construct a set of polynomial sublevel sets by

$$\mathbb{U}_k^{\mathsf{f}} := \left\{\widehat{\mathcal{U}}_{\theta,d_k} : \theta \in \Theta_k^{\mathsf{f}}\right\}.$$

By the definition of $\widetilde{\mathcal{U}}_{d_k}^{\varepsilon_k^+}$, we have $\widetilde{\mathcal{U}}_{d_k}^{\varepsilon_k^+} \in \mathbb{U}_k^{\mathsf{f}}$ for every $k \in \mathbb{N}_+$ since every $\mathbf{x} \in \mathcal{U}_{\mathsf{id}}$ also satisfies $\mathbf{x} \in \widetilde{\mathcal{U}}_{d_k}^{\varepsilon_k^+}$. As $k \to \infty$, $\widetilde{\mathcal{U}}_{d_k}^{\varepsilon_k^+}$ converges to $\widetilde{\mathcal{U}}_{d_\infty}^{\mathsf{l}} = \mathcal{U}_{\mathsf{id}}$ and thus $\mathcal{U}_{\mathsf{id}} \in \mathbb{U}_\infty^{\mathsf{f}}$.

Let $\hat{\theta}_{d_\infty}$ be one optimal solution of Problem $\mathsf{P}_{\mathsf{r},d_k}$ with $k = \infty$. Then, we continue to prove $\mathcal{U}_{\mathsf{id}} = \widehat{\mathcal{U}}_{\hat{\theta}_{d_\infty},d_\infty}$. First, we know that $\mathcal{U}_{\mathsf{id}} \subseteq \widehat{\mathcal{U}}_{\hat{\theta}_{d_\infty},d_\infty}$ due to the constraint $q(\mathbf{x}, \hat{\theta}_{d_\infty}) \leq 1, \ \forall \mathbf{x} \in \mathcal{U}_{\mathsf{id}}$. Note that we have already proved that $\mathcal{U}_{\mathsf{id}} \in \mathbb{U}_k^{\mathsf{f}}$ and let $\theta_{\mathsf{s}}$ be the parameter corresponding to the polynomial sublevel set that is identical with $\mathcal{U}_{\mathsf{id}}$. Since $\mathcal{U}_{\mathsf{id}} \subseteq \widehat{\mathcal{U}}_{\hat{\theta}_{d_\infty},d_\infty}$, we have $L(\theta_{\mathsf{s}}) \leq L(\hat{\theta}_{d_\infty})$ due to the the monotonicity of log-det inverse for positive semidefinite matrices [5]. Besides, Problem $\mathsf{P}_{\mathsf{r},d_k}$ is a convex optimization with a strictly convex objective function and thus the problem attains an unique solution. Namely, $\theta_{\mathsf{s}}$ is identical with $\hat{\theta}_{d_{infty}}$, which implies that $\mathcal{U}_{\mathsf{id}} = \widehat{\mathcal{U}}_{\hat{\theta}_{d_\infty},d_\infty}$. Since $\widehat{\mathcal{U}}_{\hat{\theta}_{d_\infty},d_\infty}$ is identical with $\widehat{\mathcal{U}}_{\theta_0^\star,d_\infty}$, (9) holds as $d \to \infty$. $\qquad \square$

**Theorem 4.** *The set $\mathcal{U}_{\mathsf{id}}$ need not to be compact. For any $\alpha > 0$, there exists a degree $d$ such that*

$$\widehat{\mathcal{U}}_{\theta_\alpha^\star,d} \subseteq \mathcal{U}_{\mathsf{id}}. \tag{15}$$

*Proof.* By Assumption 1, we know that the probability density defined on $\mathcal{U}_{\mathsf{id}}$ is positive and continuous on $\mathcal{U}_{\mathsf{id}}$.

Assume that $\theta_{\alpha,d_\infty}^\star$ is the solution of Problem $\mathsf{P}_{\alpha,d}$ with $d = d_\infty$. The corresponding polynomial sublevel set is $\widehat{\mathcal{U}}_{\theta_{\alpha,d_\infty}^\star,d_\infty}$. Note that $\widehat{\mathcal{U}}_{\theta_{\alpha,d_\infty}^\star,d_\infty}$ is compact. Define the following sets:

$$\mathcal{U}_{\mathsf{com}} := \widehat{\mathcal{U}}_{\theta_{\alpha,d_\infty}^\star,d_\infty} \bigcap \mathcal{U}_{\mathsf{id}}, \ \mathcal{U}_{\mathsf{m}} := \mathcal{U}_{\mathsf{id}} \setminus \widehat{\mathcal{U}}_{\theta_{\alpha,d_\infty}^\star,d_\infty}.$$

Assume that the volume of $\mathcal{U}_{\mathsf{m}}$ is not zero. Note that $\mathcal{U}_{\mathsf{com}}$ is compact. Replacing $\mathcal{U}_{\mathsf{id}}$ in Problem $\mathsf{P}_{\mathsf{r},d_k}$ by $\mathcal{U}_{\mathsf{com}}$, we have $\mathcal{U}_{\mathsf{com}} = \widehat{\mathcal{U}}_{\hat{\theta}_{d_\infty},d_\infty}$ by Lemma 2. Moreover, $\theta_{\alpha,d_\infty}^\star$ is a feasible solution to Problem $\mathsf{P}_{\mathsf{r},d_k}$ by $\mathcal{U}_{\mathsf{com}}$ and thus $L(\theta_{\alpha,d_\infty}^\star) > L(\hat{\theta}_{d_\infty})$ holds due to the uniqueness of the optimal solution to Problem $\mathsf{P}_{\mathsf{r},d_k}$ by $\mathcal{U}_{\mathsf{com}}$. Note that $\hat{\theta}_{d_\infty}$ is also a feasible solution of Problem $\mathsf{P}_{\alpha,d}$ with $d = d_\infty$ due to $\Pr\{\mathbf{x} \in \mathcal{U}_{\mathsf{com}}\} = 1 - \alpha$. Therefore, by $L(\theta_{\alpha,d_\infty}^\star) > L(\hat{\theta}_{d_\infty})$, it contradicts with that $\theta_{\alpha,d_\infty}^\star$ is an optimal solution. Thus, the volume of $\mathcal{U}_{\mathsf{m}}$ is zero and (15) holds. $\qquad \square$

Theorem 4 only relies on a positive value of $\alpha$ to ensure the subset relationship $\widehat{\mathcal{U}}_{\theta^{\star}_{\alpha},d} \subseteq \mathcal{U}_{\mathsf{id}}$.

**Assumption 2.** *The parameters $\alpha$ and $d$ are appropriately tuned such that Theorem 4 holds.*

**Theorem 5.** *Suppose that Assumption 2 holds. Let $\alpha_{\mathsf{c}} > \alpha$. Then, the probability that $\widehat{\mathcal{U}}_{\hat{\theta}^N_{\alpha_{\mathsf{c}}},d}$ is a subset of $\mathcal{U}_{\mathsf{id}}$ can be bounded as:*

$$\Pr\left(\widehat{\mathcal{U}}_{\hat{\theta}^N_{\alpha_{\mathsf{c}}},d} \not\subset \mathcal{U}_{\mathsf{id}}\right) \leq \exp\left(-2N^2(\alpha_{\mathsf{c}} - \alpha)\right). \tag{16}$$

*As $N \to \infty$, the bound converges to $0$.*

*Proof.* It is reasonable to assume that we seek the optimal solution of Problem $\mathsf{P}_{\alpha,d}$ within a compact set $\Theta$, which includes a solution that satisfies (12) for some $\varepsilon$ with a correspondingly chosen $d$. Besides, we also assume that $q(\mathbf{x}^{(i)}, \theta)$ is well-justified as $q(\mathbf{x}^{(i)}, \theta) = q(\mathbf{x}^{(i)}, \theta) - \varepsilon$. Let $Y_i = \mathbb{I}_1\left(q(\mathbf{x}^{(i)}, \theta)\right)$ for $i = 1, ..., N$, then $\Pr\{Y_i \in [0,1]\} = 1$ and $\mathbb{E}\{Y_i\} = \Pr\{q(\mathbf{x}, \theta) > 1\}$. Given that $\theta$ is a feasible solution for Problem $\mathsf{P}_{\alpha,d}$ and $\Pr\{q(\mathbf{x}, \theta) > 1\} \leq \alpha$. We consider the event that a feasible solution $\theta$ ($\mathbb{E}[Y_i] \leq \alpha$) for Problem $\mathsf{P}_{\alpha,d}$ is not feasible for Problem GCL, implying

$$\frac{1}{N}\sum_{i=1}^{N} \mathbb{I}_1(q\left(\mathbf{x}^{(i)}, \theta\right)) \geq \alpha_{\mathsf{c}} \Rightarrow \frac{1}{N}\sum_{i=1}^{N}(Y_i - \mathbb{E}[Y_i]) \geq \alpha_{\mathsf{c}} - \frac{1}{N}\mathbb{E}[Y_i]$$

$$\Rightarrow \frac{1}{N}\sum_{i=1}^{N}(Y_i - \mathbb{E}[Y_i]) \geq \alpha_{\mathsf{c}} - \frac{1}{N}\alpha \Rightarrow \sum_{i=1}^{N}(Y_i - \mathbb{E}[Y_i]) \geq N(\alpha_{\mathsf{c}} - \alpha).$$

Thus, we have

$$\Pr\left\{\frac{1}{N}\sum_{i=1}^{N} \mathbb{I}_1(q\left(\mathbf{x}^{(i)}, \theta\right)) \geq \alpha_{\mathsf{c}}\right\} \leq \Pr\left\{\sum_{j=1}^{N}(Y_j - \mathbb{E}[Y_j]) \geq N(\alpha_{\mathsf{c}} - \alpha)\right\} \leq \exp\{-2N(\alpha_{\mathsf{c}} - \alpha)^2\},$$

where the last inequality holds due to Hoeffding's inequality [12]. Note that $\theta^{\star}_{\alpha,d}$ is also a feasible solution of Problem $\mathsf{P}_{\alpha,d}$, which is one realization of those mentioned above $\theta$. If $\theta^{\star}_{\alpha,d}$ is a feasible solution of Problem GCL, we have that $L(\theta^{\star}_{\alpha,d}) \geq L(\hat{\theta}^N_{\alpha_{\mathsf{c}}})$. Then, $\widehat{\mathcal{U}}_{\hat{\theta}^N_{\alpha_{\mathsf{c}}},d} \subseteq \widehat{\mathcal{U}}_{\theta^{\star}_{\alpha,d}}$. By Theorem 4, we have $\widehat{\mathcal{U}}_{\hat{\theta}^N_{\alpha_{\mathsf{c}}},d} \subset \mathcal{U}_{\mathsf{id}}$. Thus, the probability of violation is bounded by $\exp\{-2N(\alpha_{\mathsf{c}} - \alpha)^2\}$. $\square$

# D   Proof of Corollary 1

From Theorem 4, we know that the inclusion $\widehat{\mathcal{U}}_{\hat{\theta}^N_{\alpha_{\mathsf{c}}},d} \subset \mathcal{U}_{\mathsf{id}}$ holds with probability at least $1 - \delta$ if the sample size satisfies $N > \sqrt{\frac{\log(1/\delta)}{2(\alpha_{\mathsf{c}} - \alpha)}}$. In addition, recall that the discount factor $\gamma$ and the OOD cost threshold $\bar{c}_{\hat{g}}$ are chosen such that the following joint chance constraint holds (Lemma 1):

$$\Pr\{\hat{g}(\mathbf{s}_h, \mathbf{a}_h) \leq 0, \ \forall h \leq H\} > 1 - \beta. \tag{17}$$

Note that the event $\left\{\widehat{\mathcal{U}}_{\hat{\theta}^N_{\alpha_{\mathsf{c}}},d} \subset \mathcal{U}_{\mathsf{id}}\right\}$ together with $\{\hat{g}(\mathbf{s}_h, \mathbf{a}_h) \leq 0, \forall h \leq H\}$ forms a sufficient condition to ensure that the agent remains within the support $\mathcal{U}_{\mathsf{id}}$ for all steps $h \leq H$.

Therefore, applying Boole's inequality yields:

$$\Pr\left\{\left[\widehat{\mathcal{U}}_{\hat{\theta}^N_{\alpha_{\mathsf{c}}},d} \not\subset \mathcal{U}_{\mathsf{id}}\right] \vee [\exists h \leq H, \ \hat{g}(\mathbf{s}_h, \mathbf{a}_h) > 0]\right\} \leq \delta + \beta. \tag{18}$$

Hence, the corollary is established: with probability at least $1 - \delta - \beta$, all state-action pairs along the trajectory remain within the support $\mathcal{U}_{\mathsf{id}}$ for the first $H$ steps. This concludes the proof of Corollary 1.

# E   Safety and Sub-Optimality with Finite Samples Considering $\diamond$'s Estimation Error

**Value Function Error.** We first analyze the error bound of the estimated value function associated with a function $\diamond$ (e.g., reward $r$ or safety cost $c_j$). Following Wachi et al. [49], we estimate $\hat{\diamond}$ using

GPR and estimate $\widehat{\mathcal{T}}$ using KDE [16]. While our theoretical analysis is grounded in these choices, our results also apply to other estimation methods, provided they ensure asymptotic consistency. Let $h$ be the bandwidth of the KDE, and assume that the joint density of $(\mathbf{s}^+, \mathbf{s}, \mathbf{a})$ and the marginal density of $(\mathbf{s}, \mathbf{a})$ are Hölder continuous with exponent $\zeta \in (0, 1]$. Let $\sigma_N(\mathbf{x})$ denote the posterior standard deviation of the GP estimate $\hat{\diamond}(\mathbf{x})$, and define $\eta_N^{1/2} := \diamond_{\mathsf{max}} + 4\omega\sqrt{\nu_N + 1 + \log(1/\delta)}$ where $\omega$ is a kernel scaling constant and $\nu_N$ is the GP information capacity. Define the maximum standard deviation at training points as: $\sigma_N^{\mathsf{max}} := \max_{\mathbf{x} \in \mathcal{U}_N} \sigma_N(\mathbf{x}), \quad \mathcal{U}_N := \left\{ \mathbf{x}^{(i)} \right\}_{i=1}^{N}$.

**Theorem 6.** *Let $\pi$ be any feasible solution of Problem* GSRL. *Assume the standard KDE conditions $Nh^{n+m} \to \infty$ and $h \to 0$ as $N \to \infty$. Then, with probability at least $1 - 2\beta - 4\delta$, the following holds:* $\left| V_{\hat{\diamond}, \widehat{\mathcal{T}}}^{\pi}(\rho_0) - V_{\diamond, \mathcal{T}}^{\pi}(\rho_0) \right| \le \varepsilon_{\mathsf{g}} + \varepsilon_{\mathsf{k}} + \varepsilon_H$, *where:*

$$\varepsilon_{\mathsf{g}} := \frac{\eta_N \sigma_N^{\mathsf{max}}}{1 - \gamma}, \ \varepsilon_H := \frac{\gamma^{H+1}(2 - \gamma)\diamond_{\mathsf{max}}}{(1 - \gamma)^2}, \ \varepsilon_{\mathsf{k}} := \frac{\diamond_{\mathsf{max}}(\gamma - \gamma^{H+2})C_{\mathsf{den}}}{(1 - \gamma)^2} \left( h^\zeta + \sqrt{\frac{\log(1/\delta)}{Nh^{2n+m}}} \right).$$

*Here, $C_{\mathsf{den}}$ is a positive constant depending on the smoothness of the densities, the choice of kernel, and the dimensionality $2n + m$.*

The proof of Theorem 6 is provided in Appendix F. By selecting a sufficiently large dataset size $N$ and a conservative OOD threshold $\bar{c}_{\hat{g}}$, we can ensure small $\beta$ in the chance constraint (4), and thus make $\varepsilon_H$ negligible. Method of choosing $\bar{c}_{\hat{g}}$ for a desired $H$ follows [50].

**Safety and Sub-optimality.** We now define conditions under which the policy output by ConOpt is safe and near-optimal with respect to the true model. We say a policy $\pi_{\mathsf{out}}$ is $\varepsilon_{\mathsf{s}}$-safe if: $\max_j \left| \bar{c}_j - V_{\hat{c}_j, \widehat{\mathcal{T}}}^{\pi_{\mathsf{out}}}(\rho_0) \right| \ge \varepsilon_{\mathsf{s}}$. Let $\hat{\pi}^*$ be the optimal solution to Problem GSRL with safety threshold $\bar{c}_j$. If $\pi_{\mathsf{out}}$ is computed using a tightened threshold $\bar{c}_j - \bar{\varepsilon}$, and satisfies: $V_{\hat{r}, \widehat{\mathcal{T}}}^{\hat{\pi}^*}(\rho_0) - V_{\hat{r}, \widehat{\mathcal{T}}}^{\pi_{\mathsf{out}}}(\rho_0) \le \varepsilon_{\mathsf{r}}$, we obtain the following guarantee for the true system:

**Theorem 7.** *If $\bar{\varepsilon} \ge \varepsilon_{\mathsf{s}} + \varepsilon_{\mathsf{g}} + \varepsilon_{\mathsf{k}} + \varepsilon_H$, and $\pi_{\mathsf{out}}$ is $\varepsilon_{\mathsf{r}}$-sub-optimal for Problem* GSRL, *then $\pi_{\mathsf{out}}$ is safe and $(\varepsilon_{\mathsf{r}} + 2\varepsilon_{\mathsf{g}} + 2\varepsilon_{\mathsf{k}} + 2\varepsilon_H)$-sub-optimal for Problem* ESRL, *with probability at least $1 - 2\beta - 4\delta$.*

# F  Proof of Theorem 6

The proof follows these main steps:

1. *Bounding Errors for Supported Policies:* We assume uniform upper bounds on the errors of conditional density estimation and reward or cost functions. Based on this, we establish the error bound for policy evaluation, limited to policies that do not visit state-action pairs outside the support of the behavior policy.

2. *Relating Sample Size and Estimation Errors:* We analyze how the sample size influences the errors in both conditional density estimation and function approximation, showing the dependency between the two.

3. *Deriving the Probabilistic Bound:* Combining the results from steps (1), (2) and Theorem 1, we deduce a probabilistic error bound for policy evaluation, demonstrating how the evaluation accuracy improves with larger datasets.

First, we give the revised telescoping Lemma by introducing the estimation error of $\hat{\diamond}$.

**Lemma 3.** *Define a function $G_{\widehat{\mathcal{T}}}^{\pi}(\mathbf{s}, \mathbf{a})$ by*

$$G_{\widehat{\mathcal{T}}}^{\pi}(\mathbf{s}, \mathbf{a}) := \mathbb{E}_{\mathbf{s}^+ \sim \widehat{\mathcal{T}}(\mathbf{s}, \mathbf{a})} \left[ V_{\diamond, \mathcal{T}}^{\pi}(\mathbf{s}^+) \right] - \mathbb{E}_{\mathbf{s}^+ \sim \mathcal{T}(\mathbf{s}, \mathbf{a})} \left[ V_{\diamond, \mathcal{T}}^{\pi}(\mathbf{s}^+) \right]. \tag{19}$$

*Then, we have*

$$V_{\hat{\diamond}, \widehat{\mathcal{T}}}^{\pi}(\rho_0) - V_{\diamond, \mathcal{T}}^{\pi}(\rho_0) = \sum_{j=0}^{\infty} \gamma^j \mathbb{E}_{\mathbf{s}_j, \mathbf{a}_j \sim \pi, \widehat{\mathcal{T}}} [\hat{\diamond}(\mathbf{s}_j, \mathbf{a}_j) - \diamond(\mathbf{s}_j, \mathbf{a}_j)] + \sum_{j=0}^{\infty} \gamma^{j+1} \mathbb{E}_{\mathbf{s}_j, \mathbf{a}_j \sim \pi, \widehat{\mathcal{T}}} \left[ G_{\widehat{\mathcal{T}}}^{\pi}(\mathbf{s}_j, \mathbf{a}_j) \right]. \tag{20}$$

*Proof.* Following the pattern of the proof of Lemma 4.1 in [55], define $W_j$ as the expected return when executing $\pi$ on $\widehat{\mathcal{M}}_{\hat{g}}$ for the $j$ steps, then switching to $\mathcal{M}$ for the remainder, written by

$$W_j = \mathop{\mathbb{E}}_{\substack{\mathbf{a}_h \in \pi(\mathbf{s}_h),\, \mathbf{s}_0 \sim \rho_0 \\ h<j:\mathbf{s}_{h+1}\sim\widehat{\mathcal{T}}(\mathbf{s}_h,\mathbf{a}_h),\, \tilde{\diamond}=\hat{\diamond} \\ h\geq j:\mathbf{s}_{h+1}\sim\mathcal{T}(\mathbf{s}_h,\mathbf{a}_h),\, \tilde{\diamond}=\diamond}} \left[ \sum_{h=0}^{\infty} \gamma^h \tilde{\diamond}(\mathbf{s}_h, \mathbf{a}_h) \right].$$

Write

$$W_j = \widehat{D}_{j-1} + \mathop{\mathbb{E}}_{\mathbf{s}_j,\mathbf{a}_j \sim \pi,\widehat{\mathcal{T}}} \left[ \gamma^j \diamond (\mathbf{s}_j, \mathbf{a}_j) + \mathop{\mathbb{E}}_{\mathbf{s}_{j+1}\sim\mathcal{T}(\mathbf{s}_j,\mathbf{a}_j)} \left[ \gamma^{j+1} V_{\diamond,\mathcal{T}}^{\pi}(\mathbf{s}_{j+1}) \right] \right]$$

$$W_{j+1} = \widehat{D}_{j-1} + \mathop{\mathbb{E}}_{\mathbf{s}_j,\mathbf{a}_j \sim \pi,\widehat{\mathcal{T}}} \left[ \gamma^j \hat{\diamond}(\mathbf{s}_j, \mathbf{a}_j) + \mathop{\mathbb{E}}_{\mathbf{s}_{j+1}\sim\widehat{\mathcal{T}}(\mathbf{s}_j,\mathbf{a}_j)} \left[ \gamma^{j+1} V_{\diamond,\mathcal{T}}^{\pi}(\mathbf{s}_{j+1}) \right] \right].$$

Here, $\widehat{D}_{j-1}$ is the expected return of the first $j-1$ time steps, which are taken with respect to $\widehat{\mathcal{T}}$ and $\hat{\diamond}$. Then, we have

$$W_{j+1} - W_j = \gamma^j \mathop{\mathbb{E}}_{\mathbf{s}_j,\mathbf{a}_j \sim \pi,\widehat{\mathcal{T}}} [\hat{\diamond}(\mathbf{s}, \mathbf{a}) - \diamond(\mathbf{s}, \mathbf{a})] + \gamma^{j+1} \mathop{\mathbb{E}}_{\mathbf{s}_j,\mathbf{a}_j \sim \pi,\widehat{\mathcal{T}}} \left[ G_{\widehat{\mathcal{T}}}^{\pi}(\mathbf{s}_j, \mathbf{a}_j) \right].$$

Note that $W_0 = V_{\diamond,\mathcal{T}}^{\pi}$ and $W_\infty = V_{\hat{\diamond},\widehat{\mathcal{T}}}^{\pi}(\rho_0)$, and we have

$$V_{\hat{\diamond},\widehat{\mathcal{T}}}^{\pi}(\rho_0) - V_{\diamond,\mathcal{T}}^{\pi}(\rho_0) = \sum_{j=0}^{\infty} (W_{j+1} - W_j)$$

$$= \sum_{j=0}^{\infty} \gamma^j \mathop{\mathbb{E}}_{\mathbf{s}_j,\mathbf{a}_j \sim \pi,\widehat{\mathcal{T}}} [\hat{\diamond}(\mathbf{s}, \mathbf{a}) - \diamond(\mathbf{s}, \mathbf{a})] + \sum_{j=0}^{\infty} \gamma^{j+1} \mathop{\mathbb{E}}_{\mathbf{s}_j,\mathbf{a}_j \sim \pi,\widehat{\mathcal{T}}} \left[ G_{\widehat{\mathcal{T}}}^{\pi}(\mathbf{s}_j, \mathbf{a}_j) \right],$$

which completes the proof. $\square$

One practical strategy is to use the kernel density estimation to give the estimations of $f(\mathbf{s}, \mathbf{x})$ and $f_{\mathbf{X}}(\mathbf{x})$ and then obtain the estimation of $p(\mathbf{s}|\mathbf{x})$. The estimation $\hat{p}(\mathbf{s}|\mathbf{x})$ is defined by

$$\hat{p}(\mathbf{s}|\mathbf{x}) = \frac{\hat{f}(\mathbf{s}, \mathbf{x})}{\hat{f}_{\mathbf{x}}(\mathbf{x})}, \tag{21}$$

where $\hat{f}(\mathbf{s}, \mathbf{x})$ and $\hat{f}_{\mathbf{x}}(\mathbf{x})$ denote the estimated joint density and marginal density, respectively. Kernel density estimation can be used for $\hat{f}(\mathbf{s}, \mathbf{x})$ and $\hat{f}_{\mathbf{x}}(\mathbf{x})$, denoting by

$$\hat{f}(\mathbf{s}, \mathbf{x}) = \frac{1}{N \cdot h^{2n+m}} \sum_{i=1}^{N} K\left(\frac{\mathbf{s} - \mathbf{s}^{(i)}}{h}\right) K\left(\frac{\mathbf{x} - \mathbf{x}^{(i)}}{h}\right), \tag{22}$$

$$\hat{f}_{\mathbf{X}}(\mathbf{x}) = \frac{1}{N \cdot h^{n+m}} \sum_{i=1}^{N} K\left(\frac{\mathbf{x} - \mathbf{x}^{(i)}}{h}\right). \tag{23}$$

We have the following lemma for the estimation error of the conditional density estimation.

**Lemma 4.** *Suppose that Assumption 1 holds. The kernel function $K(u)$ satisfies:*

$$\int K(u)\mathrm{d}u = 1,\ \sup_u |K(u)| < \infty,\ \int u^2 K(u)\mathrm{d}u < \infty. \tag{24}$$

*The bandwidth satisfies the standard kernel density estimation condition such that $Nh^{n+m} \to \infty$ and $h \to 0$ hold as $N \to \infty$. Then, let $\varepsilon_p(\mathbf{s}, \mathbf{x}) := |\hat{p}(\mathbf{s}|\mathbf{x}) - p(\mathbf{s}|\mathbf{x})|$ with probability at least $1 - \delta$, we have*

$$\varepsilon_p(\mathbf{s}, \mathbf{x}) \leq \left(\frac{C_{\mathrm{j}} + \hat{p}(\mathbf{s}|\mathbf{x}) \cdot C_{\mathrm{m}}}{f_{\mathrm{min}}}\right) \cdot \left(h^{\varsigma} + \sqrt{\frac{\log 1/\delta}{Nh^{2n+m}}}\right). \tag{25}$$

*Here, $C_{\mathrm{j}}$ is a positive constant which depends on the kernel, joint density smoothness, and dimensionality $2n + m$. Besides, $C_{\mathrm{m}}$ is a positive constant which depends on the kernel, marginal density smoothness, and dimensionality $n + m$.*

*Proof.* Compute the absolute error by

$$\hat{p}(\mathbf{s}|\mathbf{x}) - p(\mathbf{s}|\mathbf{x}) = \left| \frac{\hat{f}(\mathbf{s},\mathbf{x})}{\hat{f}_{\mathbf{x}}(\mathbf{x})} - \frac{f(\mathbf{s},\mathbf{x})}{f_{\mathbf{x}}(\mathbf{x})} \right| = \left| \frac{\hat{f}(\mathbf{s},\mathbf{x})}{\hat{f}_{\mathbf{x}}(\mathbf{x})} \cdot \frac{f_{\mathbf{x}}(\mathbf{x}) - \hat{f}_{\mathbf{x}}(\mathbf{x})}{f_{\mathbf{x}}(\mathbf{x})} - \frac{f(\mathbf{s},\mathbf{x}) - \hat{f}(\mathbf{s},\mathbf{x})}{f_{\mathbf{x}}(\mathbf{x})} \right|$$

$$\leq \frac{\left| \hat{f}(\mathbf{s},\mathbf{x}) - f(\mathbf{s},\mathbf{x}) \right|}{f_{\mathbf{x}}(\mathbf{x})} + \hat{p}(\mathbf{s}|\mathbf{x}) \cdot \frac{\left| \hat{f}_{\mathbf{x}}(\mathbf{x}) - f_{\mathbf{x}}(\mathbf{x}) \right|}{f_{\mathbf{x}}(\mathbf{x})}$$

Then, we have

$$\varepsilon_p(\mathbf{s},\mathbf{x}) \leq \underbrace{\frac{\left| \hat{f}(\mathbf{s},\mathbf{x}) - f(\mathbf{s},\mathbf{x}) \right|}{f_{\mathbf{x}}(\mathbf{x})}}_{\varepsilon_{p,1}(\mathbf{s},\mathbf{x})} + \underbrace{\hat{p}(\mathbf{s}|\mathbf{x}) \cdot \frac{\left| f_{\mathbf{x}}(\mathbf{x}) - \hat{f}_{\mathbf{x}}(\mathbf{x}) \right|}{f_{\mathbf{x}}(\mathbf{x})}}_{\varepsilon_{p,2}(\mathbf{s},\mathbf{x})}.$$

Then, According to the sup-norm bound for kernel density estimation given by Theorem 2 in [16], with probability at least $1 - \delta$, we have

$$\varepsilon_{p,1}(\mathbf{s},\mathbf{x}) \leq \frac{1}{f_{\mathsf{min}}} \cdot C_{\mathsf{j}} \cdot \left( h^{\zeta} + \sqrt{\frac{\log 1/\delta}{Nh^{2n+m}}} \right). \tag{26}$$

$$\varepsilon_{p,2}(\mathbf{s},\mathbf{x}) \leq \hat{p}(\mathbf{s}|\mathbf{x}) \cdot \frac{C_{\mathsf{m}}}{f_{\mathsf{min}}} \cdot \left( h^{\zeta} + \sqrt{\frac{\log 1/\delta}{Nh^{2n+m}}} \right). \tag{27}$$

By (26) and (27), we obtain (25). □

Based on the above discussions, we give the proof of Theorem 6 as follows.

*Proof.* (Theorem 6) We first discuss $\pi$, which ensures that $(\mathbf{s},\mathbf{a}) \sim \rho_{\mathcal{T}}^{\pi}$ stays in $\mathcal{U}_{\mathsf{id}}$ with probability 1. Using $\mathbf{x}$ for $(\mathbf{s},\mathbf{a})$, rewrite (20) into the following case:

$$\left| V_{\diamond,\widehat{\mathcal{T}}}^{\pi}(\rho_0) - V_{\diamond,\mathcal{T}}^{\pi}(\rho_0) \right| \leq \underbrace{\left| \sum_{j=0}^{\infty} \gamma^j \mathop{\mathbb{E}}_{\mathbf{x}_j \sim \pi,\widehat{\mathcal{T}}} [\hat{\diamond}(\mathbf{x}_j) - \diamond(\mathbf{x}_j)] \right|}_{\varepsilon_{\hat{\diamond}}} + \underbrace{\left| \sum_{j=0}^{\infty} \gamma^{j+1} \mathop{\mathbb{E}}_{\mathbf{x}_j \sim \pi,\widehat{\mathcal{T}}} \left[ G_{\widehat{\mathcal{T}}}^{\pi}(\mathbf{x}_j) \right] \right|}_{\varepsilon_{\widehat{\mathcal{T}}}}. \tag{28}$$

We first discuss $\varepsilon_{\hat{\diamond}}$'s bound based on the GPR-based estimation $\hat{\diamond}$. We have

$$\varepsilon_{\hat{\diamond}} = \left| \sum_{j=0}^{\infty} \gamma^j \mathop{\mathbb{E}}_{\mathbf{x}_j \sim \pi,\widehat{\mathcal{T}}} [\hat{\diamond}(\mathbf{x}_j) - \diamond(\mathbf{x}_j)] \right|$$

$$= \underbrace{\left| \sum_{j=0}^{H} \gamma^j \mathop{\mathbb{E}}_{\mathbf{x}_j \sim \pi,\widehat{\mathcal{T}}} [\hat{\diamond}(\mathbf{x}_j) - \diamond(\mathbf{x}_j)] \right|}_{\text{In distribution w.p.}(1-\beta-\delta)} + \underbrace{\left| \sum_{j=H+1}^{\infty} \gamma^j \mathop{\mathbb{E}}_{\mathbf{x}_j \sim \pi,\widehat{\mathcal{T}}} [\hat{\diamond}(\mathbf{x}_j) - \diamond(\mathbf{x}_j)] \right|}_{\text{Out of distribution}}$$

$$\leq \left| \sum_{j=0}^{H} \gamma^j \mathop{\mathbb{E}}_{\mathbf{x}_j \sim \pi,\widehat{\mathcal{T}}} [\hat{\diamond}(\mathbf{x}_j) - \diamond(\mathbf{x}_j)] \right| + \sum_{j=H+1}^{\infty} \gamma^j \diamond_{\mathsf{max}}$$

$$\leq \sum_{j=0}^{H} \gamma^j \mathop{\mathbb{E}}_{\mathbf{x}_j \sim \pi,\widehat{\mathcal{T}}} [|\hat{\diamond}(\mathbf{x}_j) - \diamond(\mathbf{x}_j)|] + \frac{\diamond_{\mathsf{max}} \cdot \gamma^{H+1}}{1-\gamma}.$$

We use the Gaussian process regression to approximate the scalar function $\diamond(\mathbf{x})$. By Theorem 6.1 of [49], we further have $\varepsilon_{\hat{\diamond}} \leq \sum_{j=0}^{H} \gamma^j \mathop{\mathbb{E}}_{\mathbf{x}_j \sim \pi,\widehat{\mathcal{T}}} [\eta_N \cdot \sigma_N(\mathbf{x}_j)] + \frac{\diamond_{\mathsf{max}} \cdot \gamma^{H+1}}{1-\gamma}$ w.p.$1 - \beta - 2\delta$. Here, $\sigma_N(\mathbf{x}_j)$ is the posterior standard deviation at point $\mathbf{x}_j$ and $\eta_N^{1/2} := \diamond_{\mathsf{max}} + 4\omega\sqrt{\nu_N + 1 + \log(1/\delta)}$ with $\omega$ as a scaling factor accounting for kernel parameters, $\nu_N$ is the information capacity associated

with kernel. Since we consider the in-distribution posterior standard deviation $\sigma_N(\mathbf{x})$, it is reasonable to assume that $\sigma_N(\mathbf{x})$ is bounded in $\mathcal{U}_s$, $\sigma_N(\mathbf{x}) \le \sigma_N^{\mathsf{max}}$. Note that $\sigma_N^{\mathsf{max}}$ can be approximately chosen as $\sigma_N^{\mathsf{max}} \approx \max_{\mathbf{x} \in \mathcal{U}_N} \sigma_N(\mathbf{x})$. Thus, we have

$$\varepsilon_{\hat{\diamond}} \le \frac{\eta_N \cdot \sigma_N^{\mathsf{max}}}{1 - \gamma} + \frac{\diamond_{\mathsf{max}} \cdot \gamma^{H+1}}{1 - \gamma} \quad \text{w.p.} 1 - \beta - 2\delta. \tag{29}$$

We then discuss the bound of $\varepsilon_{\hat{\mathcal{T}}}$. We have

$$\varepsilon_{\hat{\mathcal{T}}} = \left| \sum_{j=0}^{\infty} \gamma^{j+1} \mathop{\mathbb{E}}_{\mathbf{x}_j \sim \pi, \hat{\mathcal{T}}} \left[ G_{\hat{\mathcal{T}}}^{\pi}(\mathbf{x}_j) \right] \right| \le \underbrace{\left| \sum_{j=0}^{H} \gamma^{j+1} \mathop{\mathbb{E}}_{\mathbf{x}_j \sim \pi, \hat{\mathcal{T}}} \left[ G_{\hat{\mathcal{T}}}^{\pi}(\mathbf{x}_j) \right] \right|}_{\text{In distribution w.p.}(1 - \beta - \delta)} + \underbrace{\left| \sum_{j=H+1}^{\infty} \gamma^{j+1} \mathop{\mathbb{E}}_{\mathbf{x}_j \sim \pi, \hat{\mathcal{T}}} \left[ G_{\hat{\mathcal{T}}}^{\pi}(\mathbf{x}_j) \right] \right|}_{\text{Out of distribution}}$$

$$\le \left| \sum_{j=0}^{H} \gamma^{j+1} \mathop{\mathbb{E}}_{\mathbf{x}_j \sim \pi, \hat{\mathcal{T}}} \left[ G_{\hat{\mathcal{T}}}^{\pi}(\mathbf{x}_j) \right] \right| + \sum_{j=H+1}^{\infty} \gamma^{j+1} \cdot \frac{\diamond_{\mathsf{max}}}{1 - \gamma}$$

$$= \left| \sum_{j=0}^{H} \gamma^{j+1} \mathop{\mathbb{E}}_{\mathbf{x}_j \sim \pi, \hat{\mathcal{T}}} \left[ G_{\hat{\mathcal{T}}}^{\pi}(\mathbf{x}_j) \right] \right| + \frac{\gamma^{H+1} \cdot \diamond_{\mathsf{max}}}{(1 - \gamma)^2}$$

$$= \left| \sum_{j=0}^{H} \gamma^{j+1} \mathop{\mathbb{E}}_{\mathbf{x}_j \sim \pi, \hat{\mathcal{T}}} \left[ \mathop{\mathbb{E}}_{\mathbf{s} \sim \hat{\mathcal{T}}(\mathbf{x}_j)} \left[ V_{\diamond, \mathcal{T}}^{\pi}(\mathbf{s}) \right] - \mathop{\mathbb{E}}_{\mathbf{s} \sim \mathcal{T}(\mathbf{x}_j)} \left[ V_{\diamond, \mathcal{T}}^{\pi}(\mathbf{s}) \right] \right] \right| + \frac{\gamma^{H+1} \cdot \diamond_{\mathsf{max}}}{(1 - \gamma)^2}$$

$$= \left| \sum_{j=0}^{H} \gamma^{j+1} \mathop{\mathbb{E}}_{\mathbf{x}_j \sim \pi, \hat{\mathcal{T}}} \left[ \int_{\mathcal{S}} V_{\diamond, \mathcal{T}}^{\pi}(\mathbf{s}) \hat{p}(\mathbf{s}|\mathbf{x}_j) d\mathbf{s} - \int_{\mathcal{S}} V_{\diamond, \mathcal{T}}^{\pi}(\mathbf{s}) p(\mathbf{s}|\mathbf{x}_j) d\mathbf{s} \right] \right| + \frac{\gamma^{H+1} \cdot \diamond_{\mathsf{max}}}{(1 - \gamma)^2}$$

$$= \left| \sum_{j=0}^{H} \gamma^{j+1} \mathop{\mathbb{E}}_{\mathbf{x}_j \sim \pi, \hat{\mathcal{T}}} \left[ \int_{\mathcal{S}} V_{\diamond, \mathcal{T}}^{\pi}(\mathbf{s}) \times (\hat{p}(\mathbf{s}|\mathbf{x}_j) - p(\mathbf{s}|\mathbf{x}_j)) d\mathbf{s} \right] \right| + \frac{\gamma^{H+1} \cdot \diamond_{\mathsf{max}}}{(1 - \gamma)^2}$$

$$\le \sum_{j=0}^{H} \gamma^{j+1} \mathop{\mathbb{E}}_{\mathbf{x}_j \sim \pi, \hat{\mathcal{T}}} \left[ \int_{\mathcal{S}} \left| V_{\diamond, \mathcal{T}}^{\pi}(\mathbf{s}) \right| \times (\hat{p}(\mathbf{s}|\mathbf{x}_j) - p(\mathbf{s}|\mathbf{x}_j)) d\mathbf{s} \right] + \frac{\gamma^{H+1} \cdot \diamond_{\mathsf{max}}}{(1 - \gamma)^2}$$

$$\le \frac{\diamond_{\mathsf{max}}}{1 - \gamma} \cdot \sum_{j=0}^{H} \gamma^{j+1} \mathop{\mathbb{E}}_{\mathbf{x}_j \sim \pi, \hat{\mathcal{T}}} \left[ \int_{\mathcal{S}} \varepsilon_p(\mathbf{s}, \mathbf{x}_j) d\mathbf{s} \right] + \frac{\gamma^{H+1} \cdot \diamond_{\mathsf{max}}}{(1 - \gamma)^2}$$

(Use Lemma 4 to proceed to the next)

$$\le \frac{\diamond_{\mathsf{max}}}{1 - \gamma} \cdot \sum_{j=0}^{H} \gamma^{j+1} \mathop{\mathbb{E}}_{\mathbf{x}_j \sim \pi, \hat{\mathcal{T}}} \left[ \int_{\mathcal{S}} \left( \frac{C_{\mathsf{j}} + \hat{p}(\mathbf{s}|\mathbf{x}_j) \cdot C_{\mathsf{m}}}{f_{\mathsf{min}}} \right) \cdot \left( h^{\zeta} + \sqrt{\frac{\log 1/\delta}{Nh^{2n+m}}} \right) d\mathbf{s} \right] +$$

$$\frac{\gamma^{H+1} \cdot \diamond_{\mathsf{max}}}{(1 - \gamma)^2} \quad \text{w.p. } 1 - \beta - 2\delta$$

$$\le \frac{\diamond_{\mathsf{max}}}{1 - \gamma} \cdot \frac{\gamma(1 - \gamma^{H+1})}{1 - \gamma} \cdot \left( \frac{C_{\mathsf{j}} + C_{\mathsf{m}}}{f_{\mathsf{min}}} \right) \cdot \left( h^{\zeta} + \sqrt{\frac{\log 1/\delta}{Nh^{2n+m}}} \right) +$$

$$\frac{\gamma^{H+1} \cdot \diamond_{\mathsf{max}}}{(1 - \gamma)^2} \quad \text{w.p. } 1 - \beta - 2\delta$$

$$= \frac{\diamond_{\mathsf{max}} \cdot (\gamma - \gamma^{H+2})}{(1 - \gamma)^2} \cdot \left( \frac{C_{\mathsf{j}} + C_{\mathsf{m}}}{f_{\mathsf{min}}} \right) \cdot \left( h^{\zeta} + \sqrt{\frac{\log 1/\delta}{Nh^{2n+m}}} \right) + \frac{\gamma^{H+1} \cdot \diamond_{\mathsf{max}}}{(1 - \gamma)^2} \quad \text{w.p. } 1 - \beta - 2\delta. \tag{30}$$

Combine (29) and (30), we have that, with probability $1 - 2\beta - 4\delta$, the following holds

$$V_{\hat{\diamond},\hat{\mathcal{T}}}^{\pi}(\rho_0) - V_{\diamond,\mathcal{T}}^{\pi}(\rho_0) \leq \frac{\eta_N \cdot \sigma_N^{\max}}{1 - \gamma} + \frac{\diamond_{\max} \cdot (\gamma - \gamma^{H+2})}{(1 - \gamma)^2} \cdot C_{\mathsf{den}} \cdot \left( h^{\zeta} + \sqrt{\frac{\log 1/\delta}{Nh^{2n+m}}} \right) +$$

$$\frac{\gamma^{H+1} \cdot (2 - \gamma) \cdot \diamond_{\max}}{(1 - \gamma)^2}, \tag{31}$$

where $C_{\mathsf{den}} := (C_{\mathsf{j}} + C_{\mathsf{m}})/f_{\min}$.

$\square$

## G  Sepsis Treatment Experimental Details

### G.1  Dataset Description

The MIMIC-III (Medical Information Mart for Intensive Care) database serves as a comprehensive repository containing detailed clinical records from over $40{,}000$ intensive care admissions [18]. This extensive dataset our methods maintain clinical relevance and algorithmic robustness across diverse patient populations and treatment scenarios. Its widespread adoption in healthcare machine learning research, combined with its real-world clinical variability and detailed documentation, establishes MIMIC-III as an appropriate benchmark for treatment policy evaluation. We implemented a five-fold cross-validation approach, randomly dividing the data into training (60%), validation (20%), and test (20%) partitions for each seed.

### G.2  Sepsis Treatment Formulation for RL

The MIMIC-III Sepsis dataset provides 44 variables, comprising both dynamic physiological measurements and static patient attributes, which form the foundation for our state space construction. Following the protocols as mentioned in Section 4.1, we obtained a cohort of septic patients by identifying those who developed sepsis at some point during their ICU stay and including all observations from 24 hours before until 48 hours after the presumed onset of sepsis. The protocols organized data into 4-hour windows, creating a sequence of 20 time windows in total. Table 3 below summarizes the selected features, treatment actions, and reward signal used in our study.

Table 3: Summary of Feature Space, Actions, and Reward

| Category | Variables | Description |
|---|---|---|
| **Dynamic Features** | Mechanical ventilation, GCS, $FiO_2$, $PaO_2$, $PaO_2/FiO_2$, Total bilirubin, Urine output (4h), Cumulative urine output, Cumulative fluid input, $SpO_2$ | Time-varying physiological indicators with moderate or strong correlation to SOFA, capturing real-time sepsis severity. |
| **Static Features** | Age, Gender, Readmission status | Fixed patient characteristics offering essential context for modeling heterogeneity and enabling personalized treatment. |
| **Actions** | Intravenous fluids (per 4h), Max vasopressor dose (per 4h) | Core interventions for sepsis management aimed at stabilizing blood pressure and perfusion. |
| **Reward** | SOFA score | Quantifies the extent of organ dysfunction and guides the policy toward clinically meaningful improvement. |

**Reward Function.** The Sequential Organ Failure Assessment (SOFA) score was selected as the reward function for its clinical advantages over mortality as a terminal reward. SOFA provides instantaneous assessment of organ dysfunction across six physiological systems: respiratory, coagulation, hepatic, cardiovascular, central nervous system, and renal. SOFA scores range from 0-24, with each

subsystem contributing 0-4 points. This granularity supports more nuanced policy optimization compared to binary mortality outcomes.

**State Space.** For treatment optimization, we selected state variables through a clinically grounded and data-driven approach. Our model incorporates two treatment actions: intravenous fluid volume administered every 4 hours and maximum vasopressor dosage within the same interval. These interventions were chosen for their prevalence in early sepsis management protocols, where they restore blood pressure and tissue perfusion [33]. The state representation was designed to include features highly correlated with SOFA, capturing critical aspects of patient health relevant to clinical outcomes.

**Dynamic Feature Selection.** We prioritized dynamic variables that change during treatment by calculating Pearson correlation coefficients between each variable and the SOFA score. Both synchronous (lag = 0) and asynchronous correlations (lags of 1, 2, and 3 time steps) were examined to account for delayed physiological responses. Features with absolute correlation exceeding $0.2$ in at least one lag setting were retained. This threshold balances inclusivity and relevance—stringent enough to exclude weak associations while preserving moderately meaningful relationships to patient severity.

**Inclusion of Static Features.** To model patient heterogeneity and personalize treatment decisions, we incorporated static patient attributes that remain constant during ICU stays. Although limited to age, gender, and readmission status in this dataset, these features provide essential clinical context: age represents a known risk factor for sepsis severity, gender may influence physiological responses, and readmission could indicate chronic conditions or recent complications. Their inclusion enables policy generalization across diverse patient populations.

**Final State Space.** The final state representation comprises 13 features (10 dynamic and 3 static), creating a compact yet expressive state space that captures key clinical indicators while maintaining computational efficiency. Both state and action spaces are continuous, supporting fine-grained policy learning and clinical interpretability while remaining feasible for real-world implementation.

**Explicit Clinical Safety Constraints.** We justify that our safety oxygen saturation (SpO2) maintained at or above 92% prevents hypoxemia and ensures adequate tissue oxygenation. Urine output of at least 0.5 mL/kg/hour preserves sufficient renal perfusion and detects early signs of kidney injury. We selected these constraints to monitor distinct yet vulnerable organ systems in sepsis while providing continuous measurement capability in ICU settings, ensuring both clinical interpretability and practical implementation.

### G.3 Guardian Construction

Solving Problem GCL becomes computationally complex when the dataset is large (e.g., exceeding fifty thousand state-action pairs). To enable scalable implementation, we propose a kernel density-based approximation for the guardian set. Let $\tilde{f}_{\mathsf{pa}}(\mathbf{x}, \mathcal{X}_N)$ be a kernel density estimate built from the dataset $\mathcal{X}_N$ of state-action pairs. For any given density threshold $f_{\mathsf{ths}}$, define the corresponding empirical outlier probability by $p_{\mathsf{out}}(f_{\mathsf{ths}}) := N_{\mathsf{out}}(f_{\mathsf{ths}})/N$, where $N_{\mathsf{out}}(f_{\mathsf{ths}})$ is the number of samples in $\mathcal{X}_N$ whose estimated density is below $f_{\mathsf{ths}}$. To find a threshold corresponding to a given confidence level $\alpha$, we perform binary search over $f_{\mathsf{ths}}$: - Initialize $f_{\mathsf{ths}}^{\mathsf{min}}$ and $f_{\mathsf{ths}}^{\mathsf{max}}$ such that $p_{\mathsf{out}}(f_{\mathsf{ths}}^{\mathsf{min}}) < \alpha < p_{\mathsf{out}}(f_{\mathsf{ths}}^{\mathsf{max}})$. - Iteratively update the midpoint $f_{\mathsf{ths}}^{\mathsf{mid}} := (f_{\mathsf{ths}}^{\mathsf{min}} + f_{\mathsf{ths}}^{\mathsf{max}})/2$ and evaluate $p_{\mathsf{out}}(f_{\mathsf{ths}}^{\mathsf{mid}})$. - If $p_{\mathsf{out}}(f_{\mathsf{ths}}^{\mathsf{mid}}) > \alpha$, update $f_{\mathsf{ths}}^{\mathsf{max}} := f_{\mathsf{ths}}^{\mathsf{mid}}$; otherwise, set $f_{\mathsf{ths}}^{\mathsf{min}} := f_{\mathsf{ths}}^{\mathsf{mid}}$. After a fixed number of iterations, the binary search converges to a threshold $f_{\mathsf{ths}}^{\alpha}$ such that $p_{\mathsf{out}}(f_{\mathsf{ths}}^{\alpha}) \approx \alpha$. We then define the approximate guardian: a state-action pair $\mathbf{x}$ is considered inside the guardian if $\tilde{f}_{\mathsf{pa}}(\mathbf{x}, \mathcal{X}_N) > f_{\mathsf{ths}}^{\alpha}$, and outside otherwise.

### G.4 Policy Learning Algorithms

We explain how the guardian is applied in GCQL. CQL trains the Q-function using an offline dataset $\{(\mathbf{s}, \mathbf{a}, r, \mathbf{s}^+)\}$. During the Bellman backup step in CQL, the target is computed as $y = r + \gamma \mathbb{E}_{\mathbf{a}^+ \sim \pi(\cdot|\mathbf{s}^+)} [Q_{\mathsf{target}}(\mathbf{s}^+, \mathbf{a}^+)]$, where the expectation is approximated by sampling. Note that although $\mathbf{s}^+$ lies within the dataset, the sampled action $\mathbf{a}^+$ may result in a state-action pair $(\mathbf{s}^+, \mathbf{a}^+)$ that falls outside the guardian set. In GCQL, if $(\mathbf{s}^+, \mathbf{a}^+)$ is not within the guardian, we replace $Q_{\mathsf{target}}(\mathbf{s}^+, \mathbf{a}^+)$ with a large negative penalty value. For MB-TRPO, GMB-TRPO, MB-CPO, and GMB-CPO, we use the training data to fit a k-nearest neighbor ($k$-NN) model of the transition

dynamics. These online algorithms are then trained by interacting with the environment simulated using the $k$-NN-based transition model. The reward (i.e., the SOFA score) and the cost (i.e., $\text{SpO}_2$) are computed directly from the estimated state using predefined rules, without requiring additional model estimation. In GMB-TRPO, the guardian is incorporated by modifying the reward function: if a state-action pair falls outside the guardian set, the reward is penalized by assigning a large negative value.

### G.5 Transition Dynamics Model

To comprehensively evaluate our learned policies while maintaining safety, we employ $k$-NN model to estimate transition dynamics. The $k$-NN model learns transition dynamics from historical patient trajectories, capturing the complex relationships between medical states, treatment actions, and subsequent patient outcomes. In our experiment design, we maintain two separate $k$-NN models:

$k$-**NN-train.** Trained on the 60% training partition, used during policy learning in model-based algorithms (GMB-TRPO and GMB-CPO). As the agent learns, it takes actions in this simulated environment, with the $k$-NN model providing plausible next states based on historical patterns from the training data. This creates a realistic training environment that remains anchored to observed clinical behavior, preventing the policy from exploring dangerously unfamiliar territories.

$k$-**NN-eval.** Trained on the full dataset (training + validation + test), used exclusively for policy evaluation. This approach plays a crucial role in enabling safe off-policy evaluation by simulating patient trajectories without actual patient interaction. When evaluating a learned policy, we start with real patient states from our test set and let the policy choose treatments. The $k$-NN model then predicts the most likely next state by finding similar historical cases in the dataset. This process continues, creating synthetic patient trajectories that mirror realistic clinical progressions while keeping actual patients safe from experimental policies. The $k$-NN-eval model, having access to a broader range of state transitions, provides a more demanding test of the guardian's ability to prevent OOD exploration.

This dual-model approach ensures methodological rigor: policy learning uses only training data ($k$-NN-train), while evaluation leverages the full dataset ($k$-NN-eval) to comprehensively assess generalization. This configuration is consistent across both sepsis and hypotension experiments.

### G.6 Evaluation Metrics

**Model Concordance Rate (MCR).** The Model Concordance Rate measures the proportion of instances where the model's recommended action matches the clinician's action in the offline dataset. Formally, the MCR is defined as:

$$\text{MCR} = \frac{\sum_{i,t} \mathbb{I}\left\{\pi_{\text{SoC}}(\mathbf{s}_{i,t}) = \pi_{\text{RL}}(\mathbf{s}_{i,t})\right\}}{\sum_i T_i},$$

where $\pi_{\text{SoC}}(\mathbf{s}_{i,t})$ and $\pi_{\text{RL}}(\mathbf{s}_{i,t})$ denote the actions taken by the standard-of-care (SoC) and the learned policy at state $\mathbf{s}_{i,t}$, respectively, and $T_i$ is the number of timesteps for patient $i$. For continuous action spaces, a match is determined if the Euclidean distance between the two actions is less than a pre-specified threshold $\epsilon$, that is,

$$\|\pi_{\text{SoC}}(\mathbf{s}_{i,t}) - \pi_{\text{RL}}(\mathbf{s}_{i,t})\|_2 < \epsilon.$$

**Appropriate Intensification Rate (AIR).** The Appropriate Intensification Rate evaluates whether the model appropriately escalates treatment in response to physiological deterioration. We define the Urine Output Rate (UOR) as the volume of urine output normalized by patient weight per hour (mL/kg/hr). A need for intensification arises when either the oxygen saturation ($\text{SpO}_2$) or the UOR falls below a clinically significant threshold. Formally, AIR is defined as:

$$\text{AIR} = \frac{\sum_{i,t} \mathbb{I}\left\{\left(\text{SpO}_2(i,t) < \tau_{\text{SpO}_2} \vee \text{UOR}(i,t) < \tau_{\text{UOR}}\right) \wedge \text{Intensified}(i,t)\right\}}{\sum_{i,t} \mathbb{I}\left\{\text{SpO}_2(i,t) < \tau_{\text{SpO}_2} \vee \text{UOR}(i,t) < \tau_{\text{UOR}}\right\}},$$

where $\tau_{\text{SpO}_2}$ is set to 92%, $\tau_{\text{UOR}}$ is set to 0.5 mL/kg/hr, and $\text{Intensified}(i,t)$ is an indicator function equal to 1 if the model recommends an increased treatment intensity at time $t$.

**Mortality Estimate (ME).** The Mortality Estimate measures the likelihood of patient death under the model's policy by simulating patient trajectories using a learned transition model. Starting from an initial state, actions are selected according to the policy, and next states are predicted by the transition model. The simulation continues until either the maximum trajectory length is reached or a terminal "dead" state is encountered. The ME is defined as:

$$\text{ME} = \frac{1}{N} \sum_{i=1}^{N} \mathbb{I}\left\{\text{Diet}(i)\right\},$$

where $N$ is the number of simulated trajectories, and $\text{Diet}(i)$ equals 1 if patient $i$ equals 1 if patient enters a dead state before reaching the end of the simulation horizon.

**Action Change Penalty (ACP).** The Action Change Penalty quantifies the abruptness of the model's recommended actions across consecutive time steps within a trajectory. Formally, ACP is defined as:

$$\text{ACP} = \frac{\sum_i \sum_{t=1}^{T_i-1} \|a_{i,t+1} - a_{i,t}\|_2}{\sum_i (T_i - 1)},$$

where $a_{i,t}$ denotes the action taken at time $t$ for patient $i$, and $\|\cdot\|_2$ denotes the Euclidean norm. Lower ACP values indicate smoother and more consistent treatment recommendations over time.

### G.7 Detailed Validation Results and Discussions

#### G.7.1 Physiological State Distribution Analysis

This comprehensive analysis examines the temporal evolution of physiological states across 20-step treatment trajectories, extending beyond the safety constraints demonstrated in Figure 3 ($SpO_2$ and urine output) to encompass the broader spectrum of clinical variables. Through systematic evaluation, we reveal how different reinforcement learning implementations influence patient physiology throughout the treatment continuum.

#### G.7.2 Temporal Evolution of Clinical Variables

MB-TRPO demonstrates progressive divergence from standard care protocols, manifesting physiological deterioration across multiple organ systems illustrated in Figure 4. Mechanical ventilation patterns exhibit marked volatility, while Glasgow Coma Scale trajectories deviate substantially from clinical norms. The $PaO_2/FiO_2$ ratio reveals concerning instability that intensifies temporally, suggesting compromised respiratory efficiency. This systemic divergence indicates that unrestricted exploration permits physiologically implausible treatment strategies, compounding adverse effects across interconnected organ systems.

MB-CPO, despite explicit constraints limited to $SpO_2$ and urine output, achieves remarkable stabilization of unconstrained variables. This effect extends across multiple physiological domains: hepatic function markers demonstrate reduced variability, respiratory parameters beyond $SpO_2$ exhibit enhanced stability, and cumulative fluid balance follows more physiological trajectories. The mechanism reflects the interconnected nature of organ systems in sepsis pathophysiology, where preserved oxygenation prevents cascading organ dysfunction.

GMB-TRPO undergoes fundamental transformation, producing state distributions closely approximating standard clinical practice. This metamorphosis manifests across all monitored variables, with pronounced stabilization evident in respiratory mechanics, neurological status indicators, and fluid homeostasis parameters. The synergistic combination of explicit constraints and guardian mechanisms yields the optimal configuration, demonstrating unprecedented alignment with standard care patterns while maintaining physiological parameters within clinically appropriate ranges.

As visualized in Figure 5, MB-CQL demonstrates inherent safety properties, maintaining closer alignment with standard care practices. This alignment extends beyond action selection—where CQL achieves a MCR of 0.789 Table 1-to encompass state-space dynamics shown in Figure 1. The algorithm's conservative value function regularization naturally constrains exploration to clinically validated regions, producing physiological trajectories significantly more stable than those generated by baseline model-based approaches, such as MB-TRPO and MB-CPO. Guardian augmentation builds upon this foundation, achieving enhanced stability particularly in cumulative urine output patterns, where the combined approach maintains tighter alignment with clinical practice.

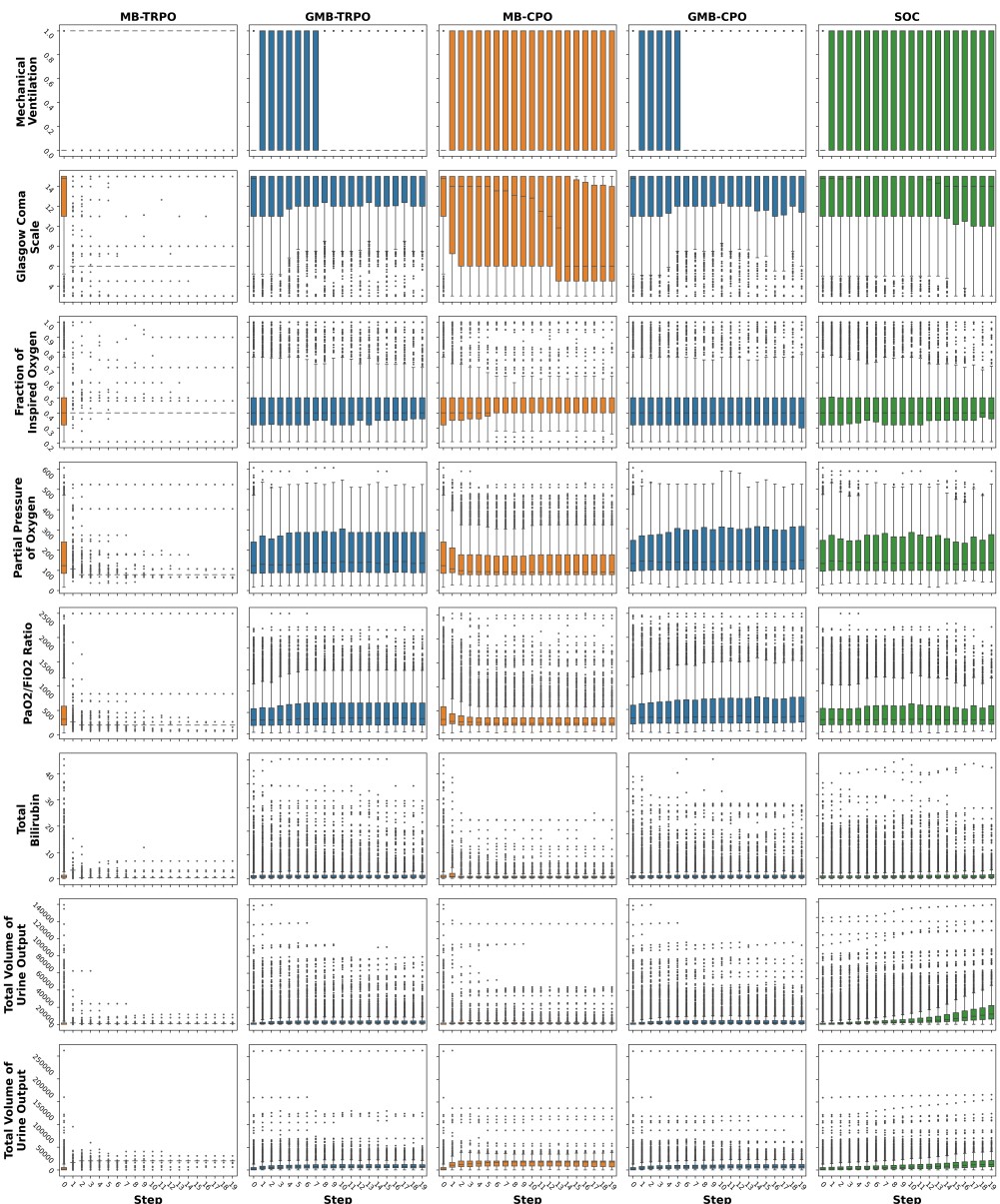

Figure 4: Physiological state progression under model-based reinforcement learning methods (MB-TRPO, GMB-TRPO, MB-CPO, and GMB-CPO). Each step is represented by a box plot, where each box shows the interquartile range (25th-75th percentiles) with the horizontal line indicating the median. Whiskers extend to $1.5 \times$IQR, and black dots represent outliers - individual measurements falling outside this range.

### G.7.3 Clinical Significance Discussion

**Physiological System Interconnectivity.** The analysis substantiates established clinical principles regarding organ system interdependence. Maintaining critical parameters within therapeutic ranges generates beneficial cascade effects throughout multiple organ systems, corroborating the therapeutic strategy of prioritizing hemodynamic stability and respiratory function as fundamental interventions in sepsis management.

**Complementary Safety Architectures.** Comparative evaluation reveals synergistic benefits between constraint-based and guardian-based protective strategies. Explicit constraints provide robust safeguards for designated variables, while guardian mechanisms furnish comprehensive trajectory

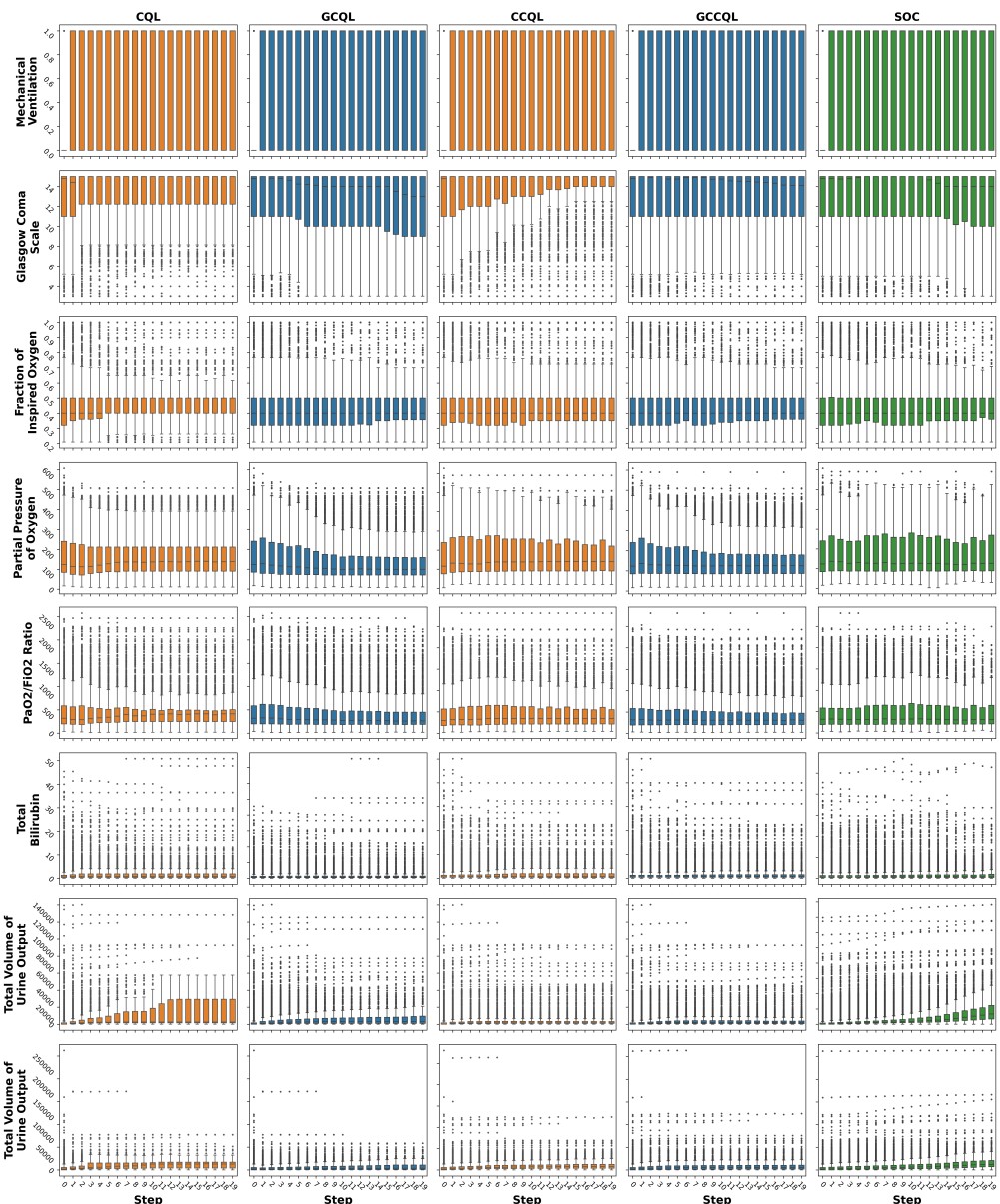

Figure 5: Physiological state progression under model-free reinforcement learning methods (CQL, GCQL, CCQL, and GCCQL). Each step is represented by a box plot, where each box shows the interquartile range (25th-75th percentiles) with the horizontal line indicating the median. Whiskers extend to 1.5×IQR, and black dots represent outliers - individual measurements falling outside this range.

protection, mitigating exploration of unsafe state combinations beyond the scope of limited constraint sets. The superior performance of GMB − CPO demonstrates that clinical implementation should incorporate both protective modalities.

**Temporal Stability Considerations.** Given sepsis pathophysiology's progressive nature, treatment protocols must maintain stability across extended periods. Guardian-enhanced methodologies exhibit superior temporal resilience, particularly crucial during the initial 20-hour therapeutic window. This consistency translates to reduced risk of abrupt physiological deterioration – a cardinal concern in intensive care settings.

**Clinical Integration Perspectives.** Alignment between guardian-enhanced policies and established care patterns facilitates integration within existing clinical workflows. Reduced variability in physiological trajectories enhances predictability, essential for clinical acceptance and real-time decision support deployment. The demonstrated capacity to identify beneficial deviations from standard protocols while maintaining safety parameters suggests potential for discovering innovative treatment approaches within established safety boundaries.

### G.7.4 Consistency Validation

**Comparison of OOD state avoidance with different seeds.**

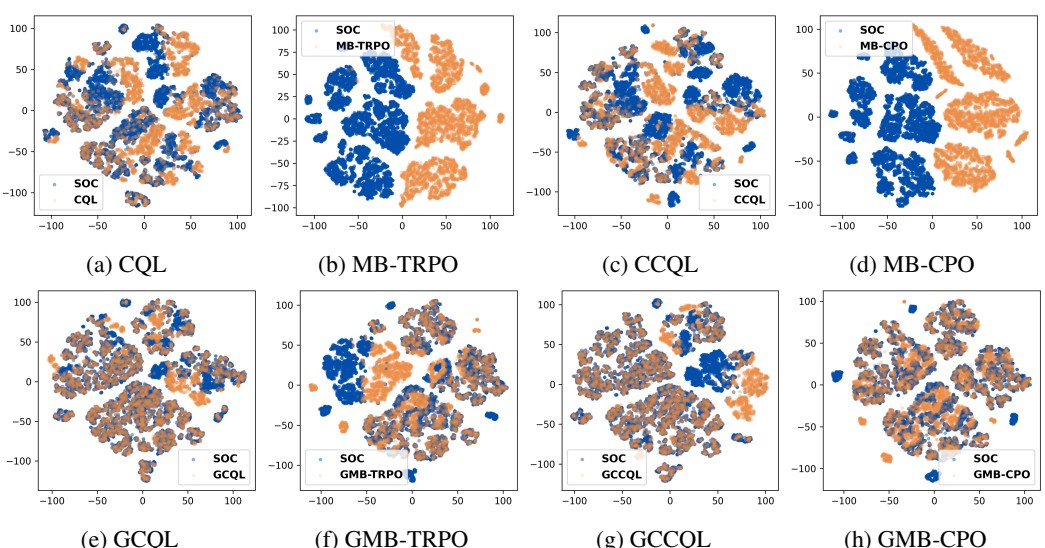

Figure 6: Results on state distributions generated using the second seed. The patterns observed here are consistent with those shown in Figure 1.

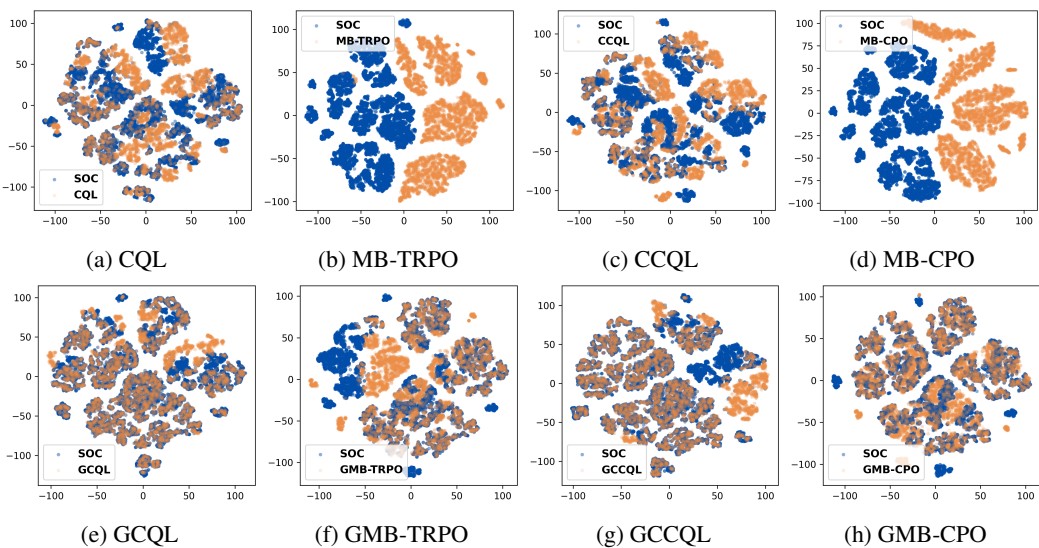

Figure 7: Results on state distributions generated using the third seed. The patterns observed here are consistent with those shown in Figure 1.

**Comparison of Cumulative Reward Distributions with Different Seeds.**

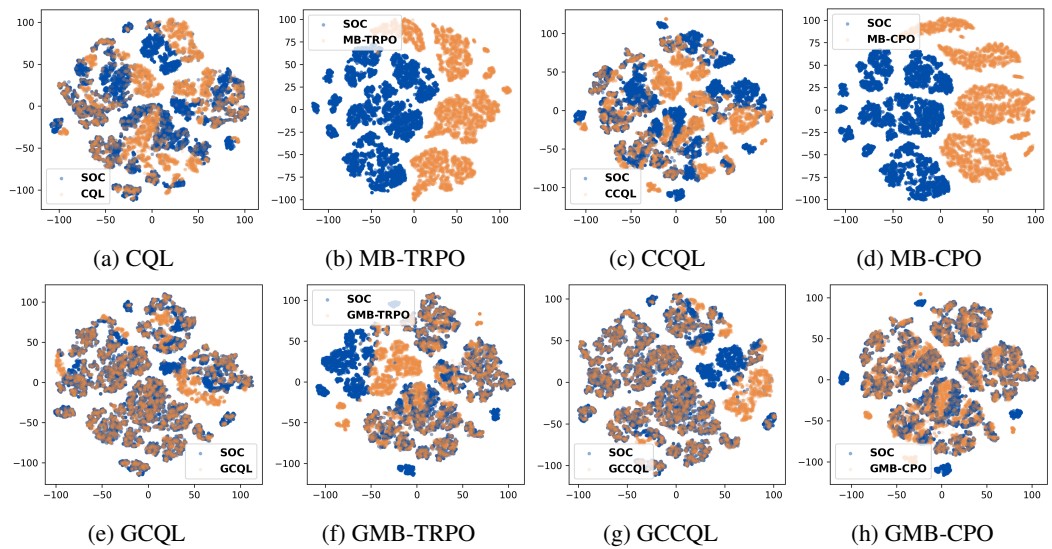

Figure 8: Results on state distributions generated using the fourth seed. The patterns observed here are consistent with those shown in Figure 1.

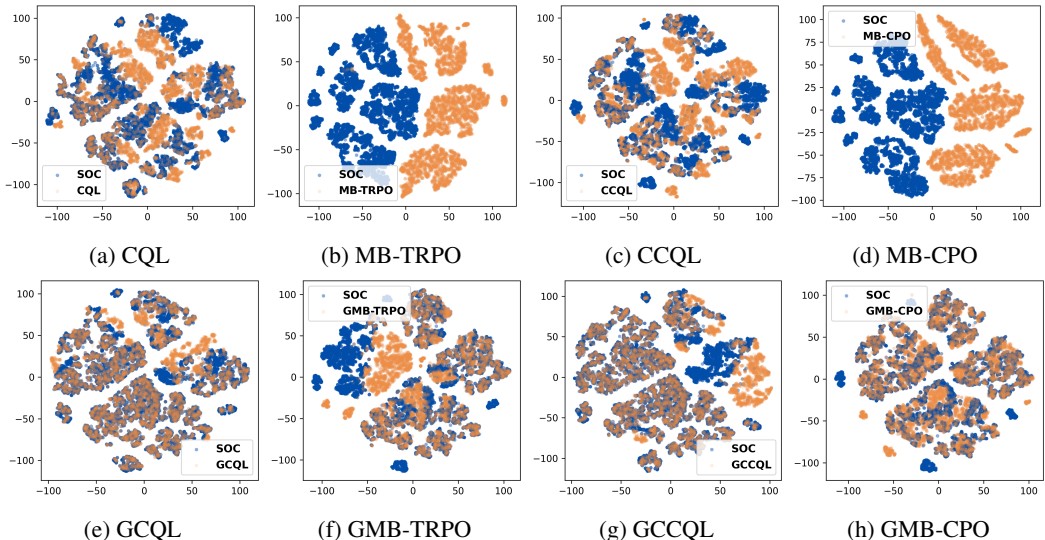

Figure 9: Results on state distributions generated using the fifth seed. The patterns observed here are consistent with those shown in Figure 1.

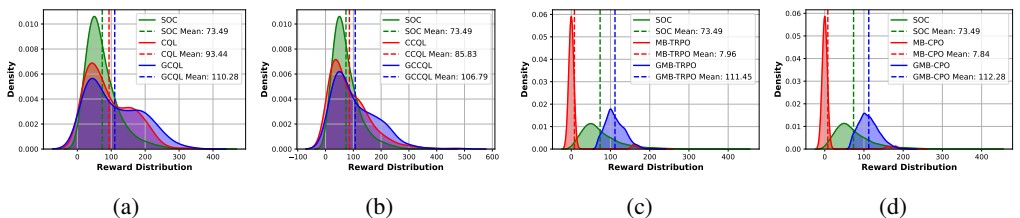

Figure 10: Comparison of cumulative reward distributions between the SOC (green) and various RL policies with guard mechanisms (blue) across different algorithms using the second seed. The patterns observed here are consistent with those shown in Figure 2. (a) CQL vs. GCQL;(b) CCQL vs. GCCQL; (c) MB-TRPO vs. GMB-TRPO; (d) MB-CPO vs. GMB-CPO.

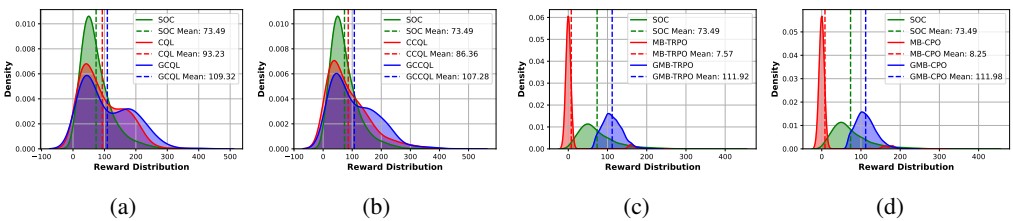

(a)  (b)  (c)  (d)

Figure 11: Comparison of cumulative reward distributions between the SOC (green) and various RL policies with guard mechanisms (blue) across different algorithms using the third seed. The patterns observed here are consistent with those shown in Figure 2. (a) CQL vs. GCQL;(b) CCQL vs. GCCQL; (c) MB-TRPO vs. GMB-TRPO; (d) MB-CPO vs. GMB-CPO.

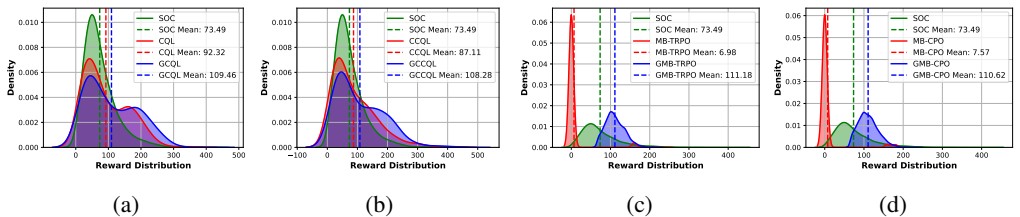

(a)  (b)  (c)  (d)

Figure 12: Comparison of cumulative reward distributions between the SOC (green) and various RL policies with guard mechanisms (blue) across different algorithms using the fourth seed. The patterns observed here are consistent with those shown in Figure 2. (a) CQL vs. GCQL;(b) CCQL vs. GCCQL; (c) MB-TRPO vs. GMB-TRPO; (d) MB-CPO vs. GMB-CPO.

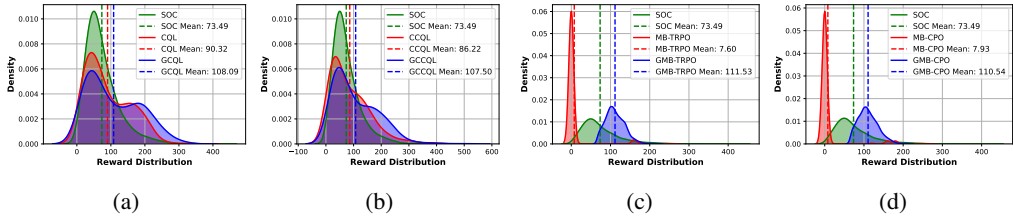

(a)  (b)  (c)  (d)

Figure 13: Comparison of cumulative reward distributions between the SOC (green) and various RL policies with guard mechanisms (blue) across different algorithms using the fifth seed. The patterns observed here are consistent with those shown in Figure 2. (a) CQL vs. GCQL;(b) CCQL vs. GCCQL; (c) MB-TRPO vs. GMB-TRPO; (d) MB-CPO vs. GMB-CPO.

### G.7.5 Comprehensive Clinical Efficacy Analysis

**Clinician Policy Alignment.** Table 1 presents a quantitative comparison of clinical alignment across methods using four metrics: MCR, AIR, ME, and ACP.

**MCR.** The alignment between policy recommendations and clinician decisions is reflected by MCR. Higher MCR values indicate stronger behavioral mimicry, which enhances clinical interpretability and acceptance. Model-free approaches such as CQL and CCQL demonstrated moderate concordance (0.789 and 0.827, respectively), while model-based methods including MB-TRPO and MB-CPO showed near-zero concordance, highlighting instability in unconstrained model-based training. The integration of OOD guardians significantly improved all model variants. GCQL achieved the highest concordance (0.909), while GMB-TRPO and GMB-CPO substantially outperformed their baseline counterparts, demonstrating the guardian's efficacy in promoting clinically familiar behavior.

**AIR.** AIR measures a learned policy's ability to intensify treatment when physiological deterioration occurs (e.g., decreased $SpO_2$ or urine output). Policies with high AIR values respond appropriately to emerging risks. Without guardian augmentation, AIR remained low across all base methods, CQL (0.130), CCQL (0.039), and MB methods ($< 0.05$)—indicating under-responsive policies. Guardian-augmented approaches substantially improved AIR, with GMB-CPO (0.448) and MB-CPO (0.496)

demonstrating the greatest responsiveness, suggesting more adaptive and clinically aligned escalation behavior.

**ME.** ME predicts expected mortality outcomes, with lower values indicating improved survival rates. Compared to standard of care (SOC: 0.0632), most baseline methods showed slight improvements (CQL: 0.0486, CCQL: 0.0481). Guardian-enhanced policies further reduced mortality estimates, with GMB-CPO achieving the lowest value (0.0138), followed by GMB-TRPO (0.0232). This suggests that guardian integration not only enhances safety but also improves health outcomes. Model-based methods without guardians produced missing or undefined mortality estimates due to trajectory instability.

**ACP.** ACP quantifies the magnitude of change in policy recommendations between consecutive time points. Lower values indicate smoother, more stable treatment suggestions—a critical property for clinical implementation, as abrupt changes in medication dosage or fluid administration can cause physiological disruption or compromise patient safety. For MVD and IFA, CQL and CCQL produced relatively stable policies (ACP of 4.18 and 3.74; ACP of $543$ and $460$, respectively). In contrast, MB-TRPO and MB-CPO exhibited substantially higher ACP values—48.1 to 50.5 for MVD and up to $4.92e^4$ for IFA—indicating erratic treatment recommendations unsuitable for clinical application.

Guardian integration dramatically improved action smoothness. GMB-CPO (ACP:MVD: $4.34$, ACP:IFA: $647$) closely matched standard-of-care values ($4.34$ and $648$), while GCQL and GCCQL maintained consistently low ACP values. These findings suggest that OOD guardians effectively regularize learned policies by discouraging unstable transitions in treatment trajectories, resulting in smoother, safer interventions that better align with clinical expectations and practices.

In summary, guardian-enhanced methodologies outperform their baseline counterparts across all metrics. Model-based approaches without guardian constraints prove unreliable due to excessive generalization, while guardian integration restores alignment with clinical norms. Among all evaluated methods, GMB-CPO delivers the most balanced performance, demonstrating strong MCR, AIR, ME, and treatment prescriptions closely resembling real-world clinical practice (ACP). These findings validate the proposed OOD guardian as an effective mechanism for ensuring safe, effective, and trustworthy policy learning in offline medical reinforcement learning.

Moreover, implicit policy (sequence of actions) adopted by clinicians might not be optimal, but individual decisions are the least safe. This is because even human experts have limited ability to integrate a patient's full historical state into consideration. Individual action is safe, but it is derived from a rigid rule or greedy fashion to achieve a short-term goal. What a capable and safe offline RL learns from observational data is a "dynamic" policy that considers the full state history of a patient with safe individual actions.

**Treatment Effectiveness and Reward Distribution.**

Figure 2 compares the cumulative reward distributions of policies learned by different RL algorithms with guardian (blue) against algorithms without guardian (red) and the standard clinical policy (green). Across all algorithms, policies trained with the guardian achieve substantially higher mean cumulative rewards than the policies trained without the guardian, highlighting the potential of the guardian to improve RL-based treatment outcomes. Notably, model-based methods such as GMB-TRPO and GMB-CPO not only yield higher reward means but also exhibit more concentrated reward distributions compared to model-free approaches (GCQL and GCCQL), demonstrating improved robustness. Furthermore, while GMB-TRPO and GMB-CPO exhibit similar performance in reward, the latter achieves lower safety costs (see Figures 3a and 3b), confirming its superior balance between reward optimization and safety compliance. Besides, for the safety cost, compared to the model-free methods GCQL and GCCQL, $GM - CPO$ shows a better similarity with the standard of care (see Figure 3). Combined with the MCR and AIR results (Tables 1), which show stronger alignment between guardian-enhanced policies and clinician decisions, these findings suggest that the OOD guardian improves not only consistency with expert behavior but also outcome quality. Among all methods, GMB-CPO achieves the best trade-off between safety and reward, addresses both OOD action and state issues, and produces robust, high-quality policies aligned with clinical practices.

**Physiological Safety.**

As shown in Figure 3, we evaluate the physiological safety of our learned treatment policies by analyzing $SpO_2$ and urine output-specifically chosen because they serve as our explicitly constrained

physiological safety states in the OGSRL framework. The results clearly demonstrate that our proposed guardian consistently reduced unsafe states compared to their non-guardian counterparts. For $SpO_2$, GCQL achieved the most substantial improvement (59.4% reduction in unsafe states), while GCCQL and GMB − CPO demonstrated strong performance with 28.8% and 49.8% reductions, respectively. Only MB − TRPO significantly worsened respiratory safety with a 90.7% increase in unsafe states, highlighting the danger of unconstrained exploration in high-stakes domains. For urine output, guardian-based methods again outperformed their counterparts, with GMB-TRPO and GMB-CPO achieving 19.0% and 19.6% reductions in unsafe states. The dual-safety mechanism in GMB-CPO, combining explicit physiological constraints with the OOD guardian, demonstrated balanced performance across both measures. Notably, even GMB-TRPO, which lacks explicit safety constraints, significantly improved safety through guardian-based restriction of OOD regions. This highlights how the guardian mechanism indirectly preserves physiological safety by constraining policies to clinically validated regions. Interestingly, we observe that model-free methods (CQL, GCQL, CCQL, GGCQL) improve $SpO_2$ safety but increase unsafe states for urine output. This pattern likely originates from $SpO_2$ responding quickly to interventions, while urine output depends on complex, delayed effects of fluid management and hemodynamic stability. Without explicit modeling of physiological dynamics, model-free methods struggle to capture these delayed treatment effects, despite successfully constraining actions to clinically observed patterns through the guardian mechanism.

## H   Acute Hypotension Experimental Details

### H.1   Dataset Description

The Synthetic Acute Hypotension Dataset [27] contains 3,910 ICU stays with 187,680 hourly state-action pairs over 48 hours. Acute hypotension (mean arterial pressure below 65 mmHg) represents a critical hemodynamic emergency distinct from sepsis's multi-organ pathophysiology. Patient-level 5-fold cross-validation follows the same methodology as sepsis: 60% training, 20% validation, 20% testing.

### H.2   Acute Hypotension Treatment Formulation for RL

**State Space.** The state space comprises 18 features: 11 continuous physiological measurements and 7 binary data availability indicators (Table 4).

Table 4: Acute hypotension state space features

| Category | Features | Type/Unit |
|---|---|---|
| Hemodynamic | MAP, Systolic BP, Diastolic BP | numeric (mmHg) |
| Respiratory | $PaO_2$, $FiO_2$ | numeric (mmHg), categorical |
| Renal | Urine output, Creatinine | numeric (mL), numeric (mg/dL) |
| Hepatic | ALT, AST | numeric (IU/L) |
| Metabolic | Lactate | numeric (mmol/L) |
| Neurological | GCS | binary |
| Measurement | Urine (M), ALT/AST (M), $FiO_2$ (M), | binary indicators |
| Indicators | GCS (M), $PaO_2$ (M), Lactate (M), Creatinine (M) | |

The 7 binary variables (with suffix (M)) indicate whether a variable was measured at a specific point in time, which in medical time series is usually highly informative. Key differences from sepsis include explicit separation of MAP, systolic, and diastolic blood pressure as primary hemodynamic indicators, hepatic function markers (ALT, AST), and metabolic marker (lactate) as a direct indicator of tissue perfusion adequacy.

**Action Space.** The action space is 2-dimensional, representing hourly fluid boluses (mL/hour) and vasopressor dosage (mcg/kg/min). While originally categorical in the dataset, we treat them as continuous to maintain consistency with sepsis experiments and enable fine-grained policy optimization.

**Reward Function.** We define a piecewise linear reward based solely on MAP:

$$r = \begin{cases} 1.0 & \text{if } \text{MAP} \geq 65 \text{ mmHg} \\ -0.2 \times (65 - \text{MAP}) & \text{if } 60 \leq \text{MAP} < 65 \text{ mmHg} \\ -1.0 - 1.0 \times (60 - \text{MAP}) & \text{if } \text{MAP} < 60 \text{ mmHg} \end{cases} \tag{32}$$

This continuous single-variable reward contrasts with the discrete multi-organ SOFA score used in sepsis. The piecewise structure ensures continuity at the breakpoint (60 mmHg) while imposing steeper penalties for critically low blood pressure. Missing MAP values are assigned zero reward. This MAP-focused reward deliberately simplifies hypotension management to test OGSRL's effectiveness across fundamentally different reward structures.

**Safety Constraints to Physiological States.** Two physiological safety costs ensure adequate organ perfusion. The urine output constraint flags states where urine production falls below 0.5 mL/kg/hour [20]. The lactate constraint flags states where lactate exceeds 2.0 mmol/L [46]. The urine threshold ensures adequate renal perfusion. The lactate threshold represents a clinically established cutoff for detecting tissue hypoperfusion—the primary pathophysiological consequence of severe hypotension. These two safety constraints provide a rigorous test of OGSRL's adaptability to disease-specific physiological boundaries.

**Guardian Construction.** KDE-based classifier is applied on 20-dimensional state-action pairs with Gaussian kernel. Complete methodology is described in Appendix G.3.

**Transition Dynamics Model.** Following the sepsis treatment methodology (Appendix G.5), we maintain two separate models: $k$-NN-train (trained on 60% training partition, used during policy learning) and $k$-NN-eval (trained on full dataset, used for policy evaluation). The $k$-NN-eval model provides a more demanding test of the guardian's ability to prevent OOD exploration by having access to a broader range of state transitions.

**Policy Learning Algorithms** To maintain conciseness while demonstrating cross-disease consistency, Table 2 reports results for representative methods: CQL and GCQL (model-free), MB-CPO and GMB-CPO (model-based), and SOC. We focus on MCR and AIR as these metrics directly assess clinical alignment and safety responsiveness—the key dimensions for validating cross-disease generalizability. Policy evaluation follows the same protocol as sepsis: sample initial states from the test set, select actions according to the learned policy, predict next states using $k$-NN-eval, and continue rollout for up to 48 steps. We compute MCR, AIR, and cumulative reward for each trajectory. All results are averaged over 5 random seeds to account for stochasticity in policy initialization and training.

## H.3 Comparison with Sepsis Experiments

Table 5: Key differences between validation domains

| Characteristic | Sepsis | Hypotension |
|---|---|---|
| Dataset source | MIMIC-III (real) | Health Gym (synthetic) |
| ICU stays | 18,923 | 3,910 |
| State-action pairs | $\sim$247,733 | 187,680 |
| Temporal resolution | 4-hour intervals | Hourly intervals |
| Episode duration | 72 hours (18 steps) | 48 hours (48 steps) |
| State dimension | 13 features | 18 features |
| Primary condition | Multi-organ dysfunction | Hemodynamic instability |
| Reward type | Discrete composite ($\text{SOFA}^{-1}$) | Continuous single-variable (MAP) |
| Safety constraints | $\text{SpO}_2$ + urine | Urine + lactate |
| Clinical objective | Reduce organ failure | Restore blood pressure |

These substantial differences test OGSRL's cross-disease generalizability across multiple dimensions: (1) *Disease pathophysiology*—sepsis involves systemic inflammation and multi-organ dysfunction, while hypotension focuses on acute hemodynamic compromise; (2) *Temporal dynamics*—hourly vs. 4-hour resolution tests adaptability to different decision frequencies; (3) *State complexity*—18 vs. 13 features tests guardian scalability; (4) *Reward structure*—discrete multi-organ composite (SOFA) vs. continuous single-variable (MAP) tests effectiveness across fundamentally different optimization

objectives; (5) *Safety constraints*—different physiological boundaries test accommodation of disease-specific requirements; (6) *Dataset size*—smaller cohort tests performance with limited data coverage. Despite these variations, consistent guardian benefits across both domains (Section 4.3) validate OGSRL as a robust framework for safe offline RL in critical care.

# I  Limitations

The OGSRL framework exhibits several constraints that merit consideration. Its conservative approach, while ensuring safety, potentially restricts the discovery of innovative treatment strategies beyond observed clinical practices—particularly relevant in evolving sepsis management. Despite advancing toward continuous representation, the implementation still simplifies the multifaceted nature of sepsis interventions, which typically encompass antibiotics, ventilation adjustments, and nutritional support beyond the modeled fluid and vasopressor dimensions. The fixed 4-hour discretization window fails to capture the rapid physiological fluctuations that might necessitate more frequent clinical interventions. Generalizability concerns arise from the MIMIC-III dataset's limited institutional scope, as treatment patterns from a single hospital system may not translate across diverse healthcare settings with varying protocols and patient demographics. Moreover, the guardian mechanism sacrifices interpretability for statistical robustness, creating potential barriers to clinical trust since its safety boundaries emerge from complex statistical properties rather than transparent medical reasoning. All these limitations are practical challenges in applications, and appropriate adaptations to the proposed OGSRL framework will be implemented for real-world clinical deployment.

# J  Experiments Compute Resources

All experiments were conducted on a high performance computing (HPC) cluster equipped with NVIDIA A100 and V100 GPUs.

# K  Broader Impact

The proposed framework, *Offline Guarded Safe Reinforcement Learning* (OGSRL), aims to improve treatment decision-making in high-stakes clinical settings using offline reinforcement learning. By introducing an OOD guardian and explicit safety cost constraints, OGSRL enables the development of safe and reliable treatment policies that remain grounded in observed clinical data. This is particularly impactful in domains such as ICU treatment, where policy optimization must adhere to strict safety boundaries due to patient risk.

The primary benefit of this work lies in its ability to learn treatment strategies that outperform clinician policies while preserving safety and trustworthiness. Since our method constrains policy learning within the support of historical clinician decisions, it ensures that learned interventions do not extrapolate dangerously beyond medical expertise. Furthermore, including theoretical safety guarantees makes our framework more suitable for deployment in clinical decision-support tools than prior offline RL approaches that lack such safeguards.

However, like all machine learning methods applied to healthcare, there are risks. Improper interpretation or deployment of learned policies without proper clinical oversight could lead to misuse. We strongly emphasize that OGSRL is designed as a decision-support tool, not a substitute for human medical judgment.

To mitigate potential negative impacts, we advocate for responsible deployment in collaboration with healthcare professionals, rigorous post-hoc evaluation in simulated environments, and continuous monitoring in real-world applications. By combining domain knowledge with safe offline learning, we believe our framework contributes positively to the development of transparent, interpretable, and trustworthy AI systems for healthcare.

