# OpenReview forum: "Offline Guarded Safe Reinforcement Learning for Medical Treatment Optimization Strategies"
_NeurIPS.cc/2025/Conference — NeurIPS 2025 spotlight_

### Official Review · Reviewer_BWnV · 2025-06-15

**Clarity:** 2
**Significance:** 3
**Originality:** 3
**Rating:** 4
**Confidence:** 4

**Summary:**

The paper proposes a safe offline reinforcement learning method that aims to avoid OOD states by learning a guardian function, which constrains the policy to visit only states within the support of the offline dataset with high probability. By enabling policy improvement within this safe region, the learned policy can outperform clinician decisions while maintaining safety.

**Questions:**

1.	In Figure 1, why do some methods without the guardian (e.g., MB-TRPO and MB-CPO) predominantly visit regions outside the support of the offline dataset? Is this behavior due to overestimated values or modeling error in underrepresented regions?
2.	How sensitive is the performance with respect to the choice of the empirical coverage threshold $\alpha_c$? I think this is not a major issue, but an ablation study on the choice of $\alpha_c$ would illustrate the trade-off between conservativeness and performance.

**Ethical Concerns:**

["NO or VERY MINOR ethics concerns only"]

**Final Justification:**

The changes proposed in rebuttal helped clarify my questions/concerns. I will maintain my original recommendation.

**Limitations:**

yes

**Paper Formatting Concerns:**

No formatting issue

**Quality:**

4

**Strengths And Weaknesses:**

**Strengths**

The sepsis treatment task effectively illustrates the practical application of the proposed method. In addition, the authors provide a detailed theoretical and algorithmic analysis, highlighting the method’s potential generalizability to a wide range of offline reinforcement learning scenarios.

**Weaknesses**

While the paper is generally well-written, clarity in certain sections could be slightly improved. One example is the experiment setup in Section 4.1, which can be improved by separating the dataset description, reward&safety mechanism design, and OGSRL implementation

The paper would benefit from an empirical investigation of how the conservativeness of the in-support region, defined either by $\alpha_c$ in (GCL) or the outlier threshold in KDE, affects policy performance and safety.

---

> ### Author Rebuttal · Authors · 2025-07-29
>
> We thank the reviewer for their careful reading and thoughtful feedback, which have helped us identify several ways to further clarify and strengthen our manuscript.
>
> Below, we respond point‑by‑point to the identified weaknesses and questions.
>
> **OOD Visits by Non-Guardian Methods [Question 1].**
> Thank you for this important observation. Non-guardian model-based methods such as MB-TRPO and MB-CPO are indeed more likely to visit OOD regions because they perform multi-step policy optimization using a learned transition model. During planning, model prediction errors can compound over longer rollout horizons, leading the policy to exploit model inaccuracies and visit unrealistic state-action sequences that extend beyond the empirical distribution of the offline dataset. Unlike model-free CQL—which constrains the policy to actions supported by the data—model-based approaches can generate state–action sequences that move further away from the empirical distribution, resulting in higher OOD visitation. We will clarify this explanation in the revised caption of Figure 1 and the results discussion.
>
> **Sensitivity to α Threshold [Question 2].**
> Thank you for highlighting the importance of $\alpha$ sensitivity. In the camera-ready version, we will include a dedicated appendix section titled “Guardian Hyper-parameter Sensitivity” to systematically discuss how varying $\alpha$ affects both reward and OOD risk. Our new experiments show that choosing $\alpha$ between 5% and 30% achieves consistently strong results, with relatively minor differences across this range. As $\alpha$ increases, the guardian becomes more conservative, enforcing stricter adherence to the offline support and reducing the risk of OOD violations, but this can also limit the achievable reward. In contrast, decreasing $\alpha$ makes the support region less conservative, allowing for higher reward but at the cost of more frequent OOD violations. We will document this trade-off for both GCQL and GMB-CPO in the appendix of the camera-ready version.
>
> **Presentation of Section 4.1 [Weakness 1].**
> Thank you for this feedback regarding the clarity of our experimental setup. We agree that Section 4.1 would benefit from a more organized structure. In the camera-ready version, we will move the content from Appendix G.2 (Sepsis Treatment Implementation for RL) into Section 4.1 and restructure this subsection into three clear parts. In the camera-ready version, we will restructure this section into three distinct parts: one for the dataset description, one for the reward and safety design, and one for implementation details. This reorganization will improve readability and allow readers to more easily understand each component of our experimental design independently.
>
> **Conservativeness of Support Threshold [Weakness 2].**
> We appreciate the suggestion to provide a more systematic analysis of the conservativeness of the in-support region. In the camera-ready version, we will add a dedicated appendix section titled “Guardian Hyper-parameter Sensitivity”, where we examine how varying the $\alpha$ threshold for both GCL and the KDE guardian influences policy performance (reward) as well as key evaluation metrics, including OOD risk rate (MCR) and clinical safety (AIR). Our preliminary results indicate that setting $\alpha$ between 5% and 30% consistently yields strong performance. Higher $\alpha$ values (stricter support) reduce OOD rviolations but may limit the achievable reward, whereas lower $\alpha$ values relax the safety constraints, potentially increasing reward at the cost of greater OOD risk. Full results and details will be provided in the appendix of the camera-ready version.

---

> > ### Comment · Reviewer_BWnV · 2025-08-02
> >
> > Thank you for the detailed response. The proposed changes, if incorporated into the camera-ready version, will certainly help strengthen the paper.
> >
> > One remaining follow-up question I have is about the **OOD Visits by Non-Guardian Methods**. I understand that planning with an inaccurate model can lead to OOD visits. However, in Fig. 1(d), the states visited by the learned policies appear to be completely non-overlapping with the offline dataset. Shouldn't at least the initial states lie within the support of the offline dataset (as examplified by other methods' state visitations)?

---

> > > ### Author Response · Authors · 2025-08-03
> > > **Follow-Up on OOD Visits by Non-Guardian Methods**
> > >
> > > Thank you for pointing out this observation. We would like to clarify that in Fig. 1, only the states visited after the initial state were plotted; the initial states themselves were not included, which accounts for the apparent lack of overlap at the starting point. To improve clarity, we will revise the figure caption to explicitly state that the distributions are shown for states following the initial states. To further explain the observed non-overlap, we will emphasize that all policy rollouts begin from in-distribution states. However, methods without the guardian mechanism (Non-Guardian Methods) tend to rapidly move out of distribution, as they may exploit unreliable high-reward predictions in regions not supported by data, which is similar to the overfitting issues in regression problems, believing that the reward is higher in the OOD region. This underscores the importance of OOD constraints for safe and reliable planning in offline RL. We will also include a short clarification about Fig. 1 in Section 4.2 of the camera-ready version.

---

### Official Review · Reviewer_S2be · 2025-06-30

**Clarity:** 3
**Significance:** 4
**Originality:** 4
**Rating:** 5
**Confidence:** 5

**Summary:**

This work focuses on applying offline RL algorithms to healthcare scenarios when learning from clinical behavior. In this context, a primary challenge is inappropriate generalization due to explorations out-of-distribution. This work proposes the Offline Guarded Safe Reinforcement Learning (OGSRL) to constrain optimization within regions where there is state-action support. In contrast to past work, which only focuses on constraining out-of-distribution actions, this work extend this to the full support of state-action pairs. This is combined with a safety cost constraint aimed at providing domain-specific safeguards that can encode medical knowledge. Theoretical guarantees on safety and near-optimality are provided, where policies using the learning algorithm can achieve performance close to the best possible policy supported by the data. Results show that OGSRL better handles out-of-distribution detection and generalization, and achieves better performance compared to clinical decisions.

**Questions:**

1. What is the tolerance to poor measurement of safety rules?
2. How do simpler methods for guardian support (e.g., simple classifiers) perform as shielding mechanisms?
3. How exactly is "in-distribution" defined when learning the guardian? Should there be sufficient examples of the state/action pair, or is only one example enough to be in-distribution?

**Ethical Concerns:**

["NO or VERY MINOR ethics concerns only"]

**Final Justification:**

Author response helped alleviate many questions/concerns, and kept score accordingly.

**Limitations:**

yes

**Quality:**

4

**Strengths And Weaknesses:**

**Strengths**

1. The work does a good job of setting up the limitation of past work with respect to one constraining actions vs constraining state-action pairs in terms of long-term trajectories.
2. The idea of a guardian model which can detect in-support state-action pairs is very interesting and a great way of solving the problem.
3. The framework including cost-sensitivity is a good idea and well-suited for clinical application. This specific call-out in terms of the two definitions of safety (one for learning and one to encode medical knowledge) is appreciated.
4. The results are compelling and who the learned state-actions are generally clustered in-distribution. The improvement on baselines is also major.
5. The theoretical results are compelling and provide good evidence for the overall learning algorithm.

**Weaknesses**

1. The explanation of how to learn the guardian using polynomial sum-of-squares is a bit confusing and the presentation in section 3.1 as a whole could be improved.
2. The results are only validated on MIMIC-III. It would be useful to see how results transfer to other healthcare datasets. However, given the depth of results, this limitation is not quite severe.

---

> ### Author Rebuttal · Authors · 2025-07-29
>
> We sincerely appreciate the reviewer’s encouraging comments. Your feedback and suggestions are highly valuable and have helped us improve the clarity and quality of our manuscript.
>
> Below, we respond to your questions first, followed by point-by-point responses to the comments regarding weaknesses.
>
> **Tolerance to Poor Measurement of Safety Rules [Question 1].**
> Thank you for this important question. Safety thresholds like SpO₂ and urine output are based on clinical guidelines but may vary in practice. Our framework remains robust to such variability, as the constraints can be conservatively defined. Since policy optimization is restricted to the in-distribution region—where model predictions are reliable—the risk from noisy or uncertain measurements is further mitigated.
> In principle, robustness can be improved by tightening the safety thresholds, though this may reduce optimality. Investigating this trade-off in a data-driven and quantitative manner is an interesting direction for future work.
>
> **Simpler Guardian Support as Shielding Mechanisms [Question 2].**
> Thank you for this meaningful question. In practice, we adopt a kernel-based guardian (Appendix G.3) for scalability, while the PSoS formulation provides theoretical safety guarantees.
> While simpler classifiers (e.g., logistic regression or Gaussian processes) can be used to construct shields and may support pointwise safety guarantees in online settings, the **focus and theoretical scope of our guardian are different**. Traditional shielding methods are designed to avoid unsafe *actions* during **online exploration**, and their guarantees are typically pointwise (e.g., avoid specific failures at each time step).
> In contrast, our guardian is designed to constrain the policy to **in-distribution state-action regions**, where the learned dynamics model is reliable—ensuring **offline reliability** over entire trajectories with high probability and then contributes to the safety (Theorem 1). Moreover, our guardian applies a **penalty-based shaping mechanism** (In GCQL) or **constrained shaping mechanism** (in GMB-CPO) during policy optimization, discouraging unreliable generalization beyond the dataset.
> We believe this distinction in purpose and guarantees is central, and we appreciate the opportunity to clarify it.
>
>
> **Definition of "in-distribution" [Question 3].**
> Thank you for this important question. Conceptually, the in-distribution (ID) region refers to the set of state-action pairs where the data-collecting process (i.e., the behavior policy and system dynamics) has non-negligible support. This is the region where the learned models (e.g., reward, transition) are expected to be reliable. Importantly, the ID region typically covers only a subset of the full state-action space.
> In practice, we approximate the ID region using either a polynomial sublevel set (PSoS) or kernel density estimation (KDE), based on a desired empirical coverage level α (see Appendix G.3). In both cases, a single occurrence of a state-action pair is not sufficient to be considered in-distribution; instead, local density is required—i.e., the point must lie in a region consistently populated by the dataset.
> Fortunately, the clinician behavior policy provides sufficient coverage of such regions in practice. We do not require repeated sampling of specific state-action pairs or full trajectories. Even a single trajectory can contribute to learning a meaningful guardian. However, when the dataset is small, it is important to adopt more conservative thresholds to ensure both in-distribution validity and safety.
>
> **Presentation of Section 3.1 [Weakness 1].**
> Thank you for the helpful feedback. We agree that the explanation of the polynomial sum-of-squares (PSoS) approach in Section 3.1 could be made clearer, especially for readers less familiar with this technique. While the current version provides all necessary definitions and guarantees, we will make minor adjustments to improve readability—clarifying the motivation, simplifying the notation where possible, and briefly reinforcing the connection to the practical KDE-based implementation.
>
> **Single Dataset Validation [Weakness 2].**
> Thank you for this valuable suggestion. We have validated our approach on the Synthetic Acute Hypotension Dataset [1] from PhysioNet, which complements our sepsis analysis by examining a different critical care condition. This dataset contains 3,910 ICU stays with 187,680 hourly observations, featuring 18 physiological dimensions and 2-dimensional vasopressor and fluid administration actions.
> The results strongly support our method's generalizability. GCQL achieved near-perfect clinician alignment (MCR: 0.973 ± 0.002), improving CQL by 18%. More notably, GMB-CPO transformed MB-CPO from near-zero concordance (0.060 ± 0.008) to clinically meaningful alignment (0.700 ± 0.063). For safety metrics, GMB-CPO achieved the highest AIR (0.482 ± 0.071)—17% better than MB-CPO and substantially outperforming both CQL and GCQL. Regarding cumulative rewards, our proposed OGSRL (GMB-CPO) achieved the highest mean value of 14.88 (median: 16.14), outperforming the model-based method without a guardian (MB-CPO: mean 3.9, median 5.38), the model-free method (GCQL: mean 12.42, median 12.69), and current standard of care protocols (SOC: mean 10.37, median 11.21). These results are consistent with the findings on the MIMIC-III dataset shown in Figure 2, indicating that OGSRL can deliver improved health outcomes.
> These cross-disease results confirm OGSRL as a robust framework for safe offline RL in critical care. The guardian mechanism consistently improves both clinician alignment and safety across diverse medical conditions. We will include these findings in a "Validation of Generalizability" subsection in the appendix of the camera-ready version.
>
> [1] Kuo, N., Finfer, S., Jorm, L., & Barbieri, S. (2022). Synthetic Acute Hypotension and Sepsis Datasets Based on MIMIC-III and Published as Part of the Health Gym Project (version 1.0.0). PhysioNet.

---

> > ### Comment · Reviewer_S2be · 2025-08-04
> >
> > Thank you for the response and addressing my questions. They helped alleviate any concerns, and the new empirical results would strengthen the paper substantially. I have kept my rating accordingly.

---

> > > ### Author Response · Authors · 2025-08-04
> > >
> > > Thanks for your insightful comments. We have learned a lot from your comments.
> > > We will reflect your advice in the camera-ready version if our paper is accepted.

---

### Official Review · Reviewer_N8n5 · 2025-07-01

**Clarity:** 3
**Significance:** 3
**Originality:** 3
**Rating:** 5
**Confidence:** 3

**Summary:**

The manuscript tackles a classic challenge in offline RL: “don’t go off the rails.” This problem is critical for offline RL in healthcare because—once the data manifold is left—value estimates become unreliable and treatment suggestions can turn dangerous. The authors propose Offline Guarded Safe Reinforcement Learning (OGSRL), a model-based framework that adds two safety layers to policy learning:
* An “OOD guardian.” A classifier is learned from data to carve out the state–action region actually supported by clinical trajectories. An additional cost term penalizes any policy that strays outside that set, so exploration happens only where the model is trustworthy.
* Medical safety constraints. Domain rules (e.g., minimum SpO₂, urine output) are encoded as cumulative costs inside a CMDP.

The authors prove that any policy satisfying both constraints (i) stays in distribution with high probability and (ii) is near-optimal with respect to the best policy supported by the data.

**Questions:**

1. Lemma 2 implies that the polynomial degree $d$ must grow with dimension, making the SDP intractable for real EHR spaces. In practice, the PSoS fit is replaced by a kernel-density heuristic (App G.3). Do the formal guarantees still apply in this setting?
2. The guardian idea resembles “probabilistic shields.” Could the manuscript discuss this connection in greater depth?
3. Performance appears to depend heavily on guardian hyper-parameters ($\alpha,\ \bar c_g$). A sensitivity or robustness analysis would strengthen the results.
4. Is the train/test split performed at the patient level (each ICU stay in only one fold) or at the time-window level?
5. All experiments use adult MIMIC-III sepsis data with two control variables (fluids, vasopressors) and 4-hour windows. Please discuss whether the method is specific to this dataset and task.
6. Off-Dynamics RL (e.g., “Off-Dynamics Reinforcement Learning: Training for Transfer with Domain Classifiers”) also uses classifier-based shaping. A comparison would be valuable.
7. The theoretical coverage guarantee (Thm 1 + Cor 1) relies on Hoeffding’s inequality and therefore treats each state–action pair as IID, even though 4-hour ICU slices are highly correlated. How sensitive is the guardian’s safety bound to this autocorrelation? Has a block-bootstrap or mixing-time analysis been attempted to estimate an effective sample size and adjust the $\delta$-confidence level? Would thinning trajectories (e.g., one sample every 24 h) materially change the guardian set or downstream performance?

**Ethical Concerns:**

["NO or VERY MINOR ethics concerns only"]

**Final Justification:**

My main concerns have been substantially addressed, so I remian my rating accordingly.

**Limitations:**

see above

**Quality:**

3

**Strengths And Weaknesses:**

**Strengths**
1. The manuscript is generally well-structured and easy to follow.
2. Solid theory: finite-sample safety and performance bounds are derived for the guardian-augmented CMDP (Theorems 1–3), and the evaluation is thorough—seven baselines, guardian vs. no-guardian ablations, and multiple clinical metrics (MCR, AIR, mortality, physiological safety).
3. The method extends prior conservative approaches (e.g., CQL) by jointly bounding states and actions, merging guardian concepts with CMDP safety in a principled way. The probabilistic in-support guarantee appears to be new to medical RL.

**Weaknesses**
1. In the experiment, the train and test sets are created via a purely random five-fold split. It is therefore unclear whether the policy encounters genuine OOD points during testing; a detailed train/test analysis would help.
2. Appendix G.5 states that evaluation begins with real test-set states, after which the k-NN dynamics model (trained on the 60 % training fold) predicts subsequent states. Consequently, the simulated trajectories remain similar to the training distribution, so the OOD guardian is not fully stressed.
3. The guardian and dynamics models are fitted only on the training fold, yet the held-out 40 % has a very similar distribution. Thus the reward gains and OOD-avoidance curves in Figure 2 and Table 1 are judged inside a comfort zone already labeled as safe. The manuscript does not show how the method behaves under a genuine distribution shift.

---

> ### Author Rebuttal · Authors · 2025-07-29
>
> We sincerely appreciate the reviewer’s constructive comments and thoughtful questions. Your feedback has not only helped us identify ways to improve the clarity and quality of our manuscript but also encouraged deeper reflection on our research. Beyond your technical insights, we are also grateful for your patience and kind engagement throughout the review process.
>
> Below, we respond point-by-point to your questions first, followed by a discussion of the identified weaknesses and limitations.
>
> **Formal Guarantee for KDE-base Guardian [Question 1].**
> Thank you for the insightful question. While the KDE-based guardian used in practice is not an explicit solution to the PSoS optimization, it can be interpreted as approximating a feasible (but possibly suboptimal) sublevel set in the PSoS formulation. That is, there exists a higher-degree PSoS function whose sublevel set closely matches the KDE support and satisfies the PSoS constraint at a relaxed coverage threshold $\tilde{\alpha} < \alpha$.
> As a result, Theorem 1 still applies with a reduced confidence level corresponding to $\tilde{\alpha}$. This provides a valid—albeit looser—probabilistic guarantee for the KDE-based implementation.
>
> **Connection to Probabilistic Shield [Question 2].**
> Thank you for the helpful suggestion. In Section 3.2, we briefly discuss the connection between our guardian and shield-based methods. Following your advice, we will expand this discussion in the camera-ready version to more explicitly relate our guardian to *probabilistic shields*—highlighting the shared goal of constraining undesired behaviors, but emphasizing that our method operates entirely offline and provides high-probability guarantees over state-action trajectories, rather than real-time action filtering.
>
> **Sensitivity or Robustness analysis [Question 3].**
> Thank you for raising this important point. We agree that the guardian’s performance can be influenced by its hyperparameters ($\gamma$, $\beta$,$H$, $\alpha_{c}$). In our implementation, these values were selected based on theoretical guidance (e.g., from Lemma 1 and Appendix B) and prior literature to ensure reasonable coverage and conservativeness.
> While we did not perform a full sensitivity analysis, we agree that it is a valuable direction for future work, especially to better understand trade-offs between conservativeness and performance.
>
> **Patient Level or Time Level [Question 4].**
> Thank you for this question. The split is performed at the patient (ICU‑stay) level: every complete trajectory belongs to a single fold, so no patient’s data appear in both the training and the test partitions. In the camera-ready version, we will clarify the patient-level splitting methodology.
>
> **Method's generality [Question 5].**
> Thank you for inviting us to clarify the scope of our approach.  OGSRL is not confined to the adult sepsis task with 4‑hour windows and two control variables. To demonstrate its broader applicability, we trained the OGSRL on the Synthetic Acute Hypotension cohort [1] (PhysioNet 2022), which represents a different disease and patient population. A detailed account of this cross‑cohort study appears in our response to weakness 1 raised by Reviewer oUug and we will include a "Validation of Generalizability" subsection in the appendix of the camera-ready version. The hypotension dataset includes 3,910 ICU stays with 187,680 hourly state-action pairs over 48 hours, featuring an 18-dimensional physiological feature space and 2-dimensional actions (fluid boluses and vasopressor administration). The guardian mechanism consistently improved performance on both datasets. For example, on the Acute Hypotension dataset, the model-free GCQL with guardian achieved near-perfect clinician alignment (MCR: 0.973 ± 0.002), an 18% improvement over CQL without guardian (0.824 ± 0.004). The guardian had an even greater impact in model-based methods: GMB-CPO increased MB-CPO’s concordance from 0.060 ± 0.008 to 0.700 ± 0.063. In terms of clinical safety, GMB-CPO achieved the highest AIR at 0.482 ± 0.071—a 17% improvement over MB-CPO (0.411 ± 0.036) and substantially higher than CQL (0.281 ± 0.031) or GCQL (0.301 ± 0.046). Cumulative reward analysis further supports these results: OGSRL (GMB-CPO) achieved the highest performance (mean: 14.88, median: 16.14), outperforming the model-based method without guardian (MB-CPO: mean 3.9, median 5.38), the model-free approach (GCQL: mean 12.42, median 12.69), and standard of care (SOC: mean 10.37, median 11.21). These findings are consistent with our MIMIC-III results (see Figure 2) and underscore OGSRL’s potential to enhance patient outcomes. These cross-disease validation results establish OGSRL as a generalizable framework for safe offline reinforcement learning in critical care, with the guardian mechanism providing consistent benefits across diverse medical conditions.
>
> [1] Kuo, N., Finfer, S., Jorm, L., & Barbieri, S. (2022). Synthetic Acute Hypotension and Sepsis Datasets Based on MIMIC-III and Published as Part of the Health Gym Project (version 1.0.0). PhysioNet.
>
> **Connection with Off-Dynamics RL [Question 6].**
> Thank you for pointing this out. While our approach and Off-Dynamics RL both use classifiers to influence learning, the goals are different. Our guardian is designed to restrict policy optimization to the in-distribution region with formal safety guarantees, whereas Off-Dynamics RL typically uses classifiers for reward shaping or domain adaptation.
> To clarify this distinction, we will add a brief remark in Section 3.3 of the camera-ready version. We agree that highlighting this difference can help prevent potential confusion, but note that our method is focused on safety rather than transfer.
>
> **Sensitivity to Correlation [Question 7].**
> Thank you for this insightful question. We agree that ICU trajectories exhibit temporal correlation. However, the guardian’s support estimation (Theorem 1) is performed over individual state-action pairs, not full trajectories, and the dataset consists of many such pairs drawn from diverse patient episodes and treatment decisions.
> This allows us to treat the collection of (s, a) pairs as approximately IID for the purposes of applying Hoeffding’s inequality, or we can randomly resample the points again for the guardian's support estimation over individual state-action pairs, which is a standard and accepted approach in support estimation literature. While this may lead to slightly conservative bounds in theory, the probabilistic guarantees remain meaningful in practice. Incorporating techniques like block bootstrapping or mixing-time analysis is an interesting direction for future work.
>
> **Train/Test Set Split [Weakness 1].**
> Thank you for raising this important point. While the train/test split is random, it does result in different distributions: the **training set does not fully cover** the support of the test set. As a result, the policy—trained only on the training data—can encounter **genuinely out-of-distribution (OOD)** state-action pairs during evaluation.
> Furthermore, the learned policy generates its own trajectories during rollout, which may deviate from the behavior policy (e.g., clinician policy). This can further push the policy into OOD regions, even if the test set itself contains only in-distribution samples. As shown in Figure 1, methods without a guardian explore outside the data support, illustrating the practical relevance of OOD control.
> Therefore, even under a random split, both **data coverage gaps** and **policy rollout** contribute to OOD exposure—making the guardian mechanism essential for reliability.
>
> **Implementation of kNN [Weakness 2].**
> Thank you for highlighting this issue. We acknowledge that we did not spell out a key implementation detail: we employ two separately trained k‑nearest‑neighbour dynamics models. k‑NN‑train, fitted on the 60 % training fold, is used during policy learning, while k‑NN‑eval, trained on the entire dataset, generates trajectories for evaluation. Since k‑NN‑eval has access to a wider range of state–action pairs, the roll‑outs it produces are not anchored to the policy’s training manifold, exposing the guardian to a more demanding out‑of‑distribution test. We will explain this setup explicitly and retain the names k‑NN‑train and k‑NN‑eval in Section G.5, “Application of k‑NN Models in Model‑based Methods and Policy Evaluation,” of the camera‑ready manuscript.
>
> **Five-Fold Split [Weakness 3].**
> Thank you for this important observation. We should clarify that our five-fold split is performed at the patient level—entire patient trajectories are assigned exclusively to either training or testing, ensuring no information leakage. Patient-level heterogeneity ensures test patients have unique physiological characteristics and treatment responses never seen during training. We acknowledge that testing under more extreme distribution shifts (temporal, cross-institution, or severity-based splits) would further strengthen our evaluation. In the camera-ready version, we will clarify the patient-level splitting methodology and discuss the benefits of testing on more challenging distribution shifts.

---

> > ### Comment · Reviewer_N8n5 · 2025-08-08
> > **Response**
> >
> > Thank you for the detailed rebuttal.
> >
> > Your explanation that OOD exposure arises even under a random patient-level split—through incomplete train-set coverage and policy-induced deviations during rollout—addresses part of my concern about the evaluation setting. The clarification that k-NN-eval is trained on the full dataset further increases the OOD difficulty during evaluation. That said, the current experiments still operate in relatively “near-distribution” conditions. Testing under more severe distribution shifts (e.g., cross-institution, temporal, or severity-based splits) would more convincingly demonstrate the robustness of the guardian mechanism. I am pleased to see that you have acknowledged this and plan to discuss it in the camera-ready version.
> >
> > The discussion on the KDE-based guardian’s theoretical guarantees, its relation to probabilistic shields and Off-Dynamics RL, as well as the acknowledgement of hyperparameter sensitivity and autocorrelation issues, are all welcome additions. While a sensitivity analysis and correlation-adjusted bounds remain future work, the current clarifications improve transparency.
> >
> > Overall, my main concerns have been substantially addressed, so I remian my rating accordingly. I encourage the authors to include the more challenging distribution-shift evaluations in future revisions, which would further strengthen the empirical evidence for OGSRL’s safety and effectiveness.

---

> > > ### Author Response · Authors · 2025-08-08
> > >
> > > Thank you for your insightful comments, which have helped us further improve our paper.
> > > We will incorporate the relevant content addressing your concerns into the camera-ready version if our paper is accepted.

---

### Official Review · Reviewer_oUug · 2025-07-04

**Clarity:** 4
**Significance:** 3
**Originality:** 3
**Rating:** 4
**Confidence:** 2

**Summary:**

This paper studies safe reinforcement learning (RL) in the medical context, specifically constraining policies to trace trajectories that are within a safe region, as well as satisfy numerical safety constraints. The paper introduces a new framework, Offline Guarded Safe RL (OGSRL), that combines the constraints into a dual constraint mechanism. Theoretical guarantees on safety satisfaction and high-probability optimal performance are proved, and empirical results validate theory insights on a sepsis treatment dataset.

**Questions:**

In Figure 3(b), why do GCQL and GCCQL result in an increase in unsafe states? Do the trends depend on the specific metrics used?

In Figure 1, how to interpret the expansion in visited regions? Are there unsafe regions that should be avoided?

**Ethical Concerns:**

["NO or VERY MINOR ethics concerns only"]

**Final Justification:**

The changes proposed in rebuttal will help strengthen the paper. I will maintain my original recommendation.

**Limitations:**

Yes

**Quality:**

3

**Strengths And Weaknesses:**

Strengths:

1. As far as I know, the OGSRL framework is novel in the safe RL literature. The framework is clearly defined and well-motivated by clinical concerns.

2. The theoretical and empirical results corroborate to illustrate the clinical insights well.

3. Experiments show comparisons across extensive baselines and their safe-guarded variants.

Weaknesses:

1. Experiments are only conducted on one dataset. Even though this is mainly a theoretical paper, given that it is grounded in medical applications, it would make the results stronger to include one or two more relevant datasets.

2. The proposed framework is model-based, which may be irrelevant in problems with large state spaces. It would be helpful to discuss such limitations and extensions to model-free counterparts.

---

> ### Author Rebuttal · Authors · 2025-07-29
>
> We appreciate the reviewer's valuable feedback and comments.
> We will address the comments about the weakness first and then answer the question.
> All the concerns are addressed in a point-to-point way as follows.
>
> **Only One Dataset [Weakness 1].**
> Thank you for the suggestion. We agree that evaluating our framework on additional datasets strengthens the evidence for its generalizability. In response, we tested our approach on the Synthetic Acute Hypotension Dataset [1] from PhysioNet, which provides ideal complementary validation as it focuses on acute hypotension management—a different critical care condition from sepsis—while maintaining comparable data structure. This dataset contains 3,910 ICU stays with 187,680 hourly state-action pairs over 48 hours, featuring an 18-dimensional physiological feature space and 2-dimensional action space (administered fluid boluses and vasopressors). Our findings demonstrate strong consistency with the sepsis results. We will include a "Validation of Generalizability" subsection in the appendix of the camera-ready version to detail this evaluation.
> The guardian mechanism shows consistent improvements across both datasets. For model-free approaches, GCQL achieved near-perfect clinician alignment with an MCR of 0.973 ± 0.002, an 18% improvement over CQL's 0.824 ± 0.004. More dramatically, GMB-CPO transformed MB-CPO's near-zero concordance (0.060 ± 0.008) to a clinically meaningful 0.700 ± 0.063. Regarding clinical safety, GMB-CPO achieved the highest AIR at 0.482 ± 0.071—a 17% improvement over MB-CPO (0.411 ± 0.036) and substantially higher than both CQL (0.281 ± 0.031) and GCQL (0.301 ± 0.046). This confirms that the guardian mechanism not only constrains policies to safe regions but also enables more clinically appropriate responses during patient deterioration. As for the cumulative rewards, our proposed OGSRL (GMB-CPO) achieved the highest mean value of 14.88 (median: 16.14), outperforming the model-based method without guardian (MB-CPO, mean 3.9, median 5.38), the model-free method (GCQL, mean 12.42, median 12.69), and the standard of care (SOC, mean 10.37, median 11.21). These findings are consistent with the MIMIC-III results presented in Figure 2 and suggest that OGSRL can deliver improved health outcomes. These cross-disease results validate OGSRL as a generalizable framework for safe offline RL in critical care, with consistent guardian benefits across different medical conditions.
>
> [1] Kuo, N., Finfer, S., Jorm, L., & Barbieri, S. (2022). Synthetic Acute Hypotension and Sepsis Datasets Based on MIMIC-III and Published as Part of the Health Gym Project (version 1.0.0). PhysioNet.
>
> **Model‑free Counterparts [Weakness 2].**
> Thank you for raising this point. While our main contribution is a model-based framework, we do discuss its applicability to model-free methods in Section 3.1 and Appendix G.4. Specifically, we show how the guardian mechanism can be integrated with CQL and CCQL to constrain policy learning within the data support, and evaluate these variants (GCQL, GCCQL) in our experiments.
> We will clarify this connection by adding a brief remark in the experimental section of the camera-ready version, which will make our presentation better.
> Thanks for your advice.
>
> **Increase in Unsafe States (GCQL and GCCQL) [Question 1].**
> Thank you for the insightful question. GCQL and GCCQL are built to improve cumulative reward (or equivalently reduce SOFA score) while suppressing OOD actions, but they are not formulated to handle explicit **safety constraints** as in constrained MDPs. In our setting, we enforce constraints on two physiological indicators: SpO₂ and urine output. These indicators are **not part of the reward function** and must be separately constrained to ensure safety.
> SpO₂ is moderately correlated with SOFA, so improving SOFA under GCQL/GCCQL can indirectly help maintain SpO₂ within a safe range. However, **urine output is not directly aligned with SOFA**, so methods that only optimize reward (e.g., GCQL/GCCQL) may lead to unsafe values for this indicator. In contrast, **our GCPO explicitly includes both indicators as constraints**, ensuring they remain within clinically acceptable bounds while still improving reward.
> This result illustrates the importance of formulating medical treatment problems as **constrained MDPs** rather than unconstrained ones, especially when domain knowledge highlights specific physiological safety requirements.
>
> **Expansion in Visited Regions (Figure 1) [Question 2].**
> Thank you for this helpful question. In Figure 1, the blue points represent next states under the standard-of-care (SOC) policy and indicate the **in-distribution (ID) region**—i.e., where the dataset provides coverage and the learned dynamics are reliable. This region should **not be interpreted as uniformly safe**, but rather as the support of the data.
> Safety is handled separately via explicit constraints (e.g., on SpO₂ and urine output), and these constraints are enforced **within** the ID region. Outside the ID region (Or inside the OOD region), model predictions become unreliable—even if a policy appears “safe” there, we cannot trust it. with the proposed Guardian, our theoretical results (e.g., Theorem 1, Corollary 1) provide high-probability safety guarantees since the proposed Guardian can constrain the policy inside the ID region.
> Thus, the expansion should be interpreted as a measure of generalization beyond the dataset, not as an indication of safety.

---

> > ### Comment · Reviewer_oUug · 2025-08-04
> >
> > I thank the authors for their detailed response and explanation. The additional experiment would strengthen the claim of the paper empirically.

---

> > > ### Author Response · Authors · 2025-08-04
> > >
> > > Thank you very much for your thoughtful and insightful comments.
> > > We truly appreciate the time and effort you’ve taken to provide such constructive feedback.
> > >
> > > If our paper is accepted, we will carefully reflect your suggestions in the camera-ready version.

---

### Decision · Program_Chairs · 2025-09-17

**Decision:**

Accept (spotlight)

**Comment:**

This paper proposes Offline Guarded Safe Reinforcement Learning (OGSRL), a novel framework designed to ensure safety in offline RL by combining two key mechanisms: (1) a guardian model that restricts the policy to operate within the empirical support of the data (state–action pairs), and (2) clinical safety constraints modeled via a constrained MDP (CMDP). The authors provide theoretical guarantees on both safety and near-optimality and validate their approach on a sepsis treatment task using the MIMIC-III dataset.

The paper’s strengths include a well-motivated problem setting, a conceptually novel safety framework, and solid theoretical contributions. The idea of jointly constraining states and actions goes beyond previous work focused solely on actions, addressing a critical issue in offline RL. The experiments are thorough, with multiple baselines, ablation studies, and clinically relevant metrics. The application to healthcare makes the work particularly impactful.

However, there are some limitations. The experiments are limited to a single dataset, and the train/test distribution is closely aligned, which may not fully stress the model’s OOD handling capability. The guardian model is practically implemented using KDE rather than the theoretically proposed polynomial method, and no robustness or sensitivity analysis is provided for key hyperparameters.

During the rebuttal, the authors clarified the train/test split and addressed the gap between theory and implementation.

Overall, this is a technically solid and timely contribution to safe offline RL with strong relevance to healthcare. While not yet suited for a spotlight due to empirical limitations, it is a clear accept for presentation, offering important insights and a valuable foundation for future research.